# On Margin Maximization in Linear and ReLU Networks

**Gal Vardi**
TTI-Chicago and Hebrew University[*]
galvardi@ttic.edu

**Ohad Shamir**
Weizmann Institute of Science
ohad.shamir@weizmann.ac.il

**Nathan Srebro**
TTI-Chicago
nati@ttic.edu

## Abstract

The implicit bias of neural networks has been extensively studied in recent years. Lyu and Li [2019] showed that in homogeneous networks trained with the exponential or the logistic loss, gradient flow converges to a KKT point of the max margin problem in parameter space. However, that leaves open the question of whether this point will generally be an actual optimum of the max margin problem. In this paper, we study this question in detail, for several neural network architectures involving linear and ReLU activations. Perhaps surprisingly, we show that in many cases, the KKT point is not even a *local* optimum of the max margin problem. On the flip side, we identify multiple settings where a local or global optimum can be guaranteed.

## 1 Introduction

A central question in the theory of deep learning is how neural networks generalize even when trained without any explicit regularization, and when there are far more learnable parameters than training examples. In such optimization problems there are many solutions that label the training data correctly, and gradient descent seems to prefer solutions that generalize well [Zhang et al., 2016]. Hence, it is believed that gradient descent induces an *implicit bias* [Neyshabur et al., 2014, 2017], and characterizing this bias has been a subject of extensive research in recent years.

A main focus in the theoretical study of implicit bias is on *homogeneous* neural networks. These are networks where scaling the parameters by any factor $\alpha > 0$ scales the predictions by $\alpha^L$ for some constant $L$. For example, fully-connected and convolutional ReLU networks without bias terms are homogeneous. Lyu and Li [2019] proved that in linear and ReLU homogeneous networks trained with the exponential or the logistic loss, if gradient flow (GF) converges to a sufficiently small loss[2], then the direction to which the parameters of the network converge can be characterized as a first order stationary point (KKT point) of the maximum margin problem in parameter space. Namely, the problem of minimizing the $\ell_2$ norm of the parameters under the constraints that each training example is classified correctly with margin at least $1$. They also showed that this KKT point satisfies necessary conditions for optimality. However, the conditions are not known to be sufficient even for local optimality. It is analogous to showing that some unconstrained optimization problem converges to a point with gradient zero, without proving that it is either a global or a local minimum. Thus, the

---

[*]Work done while the author was at the Weizmann Institute of Science
[2]They also assumed directional convergence, but [Ji and Telgarsky, 2020] later showed that this assumption is not required.

36th Conference on Neural Information Processing Systems (NeurIPS 2022).

Table 1: Results on depth-2 networks

|  | Linear | ReLU |
|---|---|---|
| Fully-connected | Global (Thm. 3.1) | Not local (Thm. 3.2) |
| $\mathcal{N}_{\text{no-share}}$ | Not local (Thm. 4.1) | Not local (Thm. 3.2) |
| $\mathcal{N}_{\text{no-share}}$ assuming non-zero weights vectors | Global (Thm. 4.2) | Not local (Thm. 4.2) |
| $\mathcal{N}_{\text{no-share}}$ assuming non-zero inputs to all neurons | Global (Thm. 4.2) | Local, Not global (Thm. 4.3) |
| $\mathcal{N}$ assuming non-zero inputs to all neurons | Not local (Thm. 4.4) | Not local (Thm. 4.4) |

question of when GF maximizes the margin remains open. Understanding margin maximization may be crucial for explaining generalization in deep learning, and it might allow us to utilize margin-based generalization bounds for neural networks.

In this work we consider several architectures of homogeneous neural networks with linear and ReLU activations, and study whether the aforementioned KKT point is guaranteed to be a global optimum of the maximum margin problem, a local optimum, or neither. Perhaps surprisingly, our results imply that in many cases, such as depth-2 fully-connected ReLU networks and depth-2 diagonal linear networks, the KKT point may not even be a *local* optimum of the maximum-margin problem. On the flip side, we identify multiple settings where a local or global optimum can be guaranteed.

We now describe our results in a bit more detail. We denote by $\mathcal{N}$ the class of neural networks without bias terms, where the weights in each layer might have an arbitrary sparsity pattern, and weights might be shared[3]. The class $\mathcal{N}$ contains, for example, *convolutional networks*. Moreover, we denote by $\mathcal{N}_{\text{no-share}}$ the subclass of $\mathcal{N}$ that contains only networks without shared weights, such as *fully-connected networks* and *diagonal networks* (cf. Gunasekar et al. [2018b], Yun et al. [2020]). We describe our main results below, and also summarize them in Tables 1 and 2.

**Fully-connected networks:**

- In linear fully-connected networks of any depth the KKT point is a global optimum[4].

- In fully-connected depth-2 ReLU networks the KKT point may not even be a local optimum. Moreover, this negative result holds with constant probability over the initialization, i.e., there is a training dataset such that GF with random initialization converges with positive probability to the direction of a KKT point which is not a local optimum.

**Depth-2 networks in $\mathcal{N}$:**

- In linear networks with sparse weights, and specifically in diagonal networks, we show that the KKT point may not be a local optimum.

- In our proof of the above negative result, the KKT point contains a neuron whose weights vector is zero. However, in practice gradient descent often converges to networks that do not contain such zero neurons. We show that for linear networks in $\mathcal{N}_{\text{no-share}}$, if the KKT point has only non-zero weights vectors, then it is a global optimum. Thus, despite the above negative result, a reasonable assumption on the KKT point allows us to obtain a strong positive result. We also show some implications of our results on margin maximization in predictor space for depth-2 diagonal linear networks (see Remark 4.1).

- For ReLU networks in $\mathcal{N}_{\text{no-share}}$, in order to obtain a positive result we need a stronger assumption. We show that if the KKT point is such that for every input in the dataset the input to every hidden

---

[3]See Section 2 for the formal definition.

[4]We note that margin maximization for such networks in predictor space is already known [Ji and Telgarsky, 2020]. However, margin maximization in predictor space does not necessarily imply margin maximization in parameter space.

Table 2: Results on deep networks

|  | Linear | ReLU |
|---|---|---|
| Fully-connected | Global (Thm. 3.1) | Not local (Thm. 3.2) |
| $\mathcal{N}_{\text{no-share}}$ assuming non-zero inputs to all neurons | Not local (Thm. 5.1) | Not local (Thm. 5.1) |
| $\mathcal{N}$ - max margin for each layer separately | Global (Thm. 5.2) | Not local (Thm. 5.3) |
| $\mathcal{N}$ - max margin for each layer separately, assuming non-zero inputs to all neurons | Global (Thm. 5.2) | Local, Not global (Thm. 5.4) |

neuron in the network is non-zero, then it is guaranteed to be a local optimum (but not necessarily a global optimum).

- We prove that assuming the network does not have shared weights is indeed required in the above positive results, since for networks with shared weights (such as convolutional networks) they no longer hold.

**Deep networks in $\mathcal{N}$:**

- We discuss the difficulty in extending our positive results to deeper networks. Then, we study a weaker notion of margin maximization: maximizing the margin for each layer separately. For linear networks of depth $m \geq 2$ in $\mathcal{N}$ (including networks with shared weights), we show that the KKT point is a global optimum of the per-layer maximum margin problem. For ReLU networks the KKT point may not even be a local optimum of this problem, but under the assumption on non-zero inputs to all neurons it is a local optimum.

As detailed above, we consider several different settings, and the results vary dramatically between the settings. Thus, our results draw a somewhat complicated picture. Overall, our negative results show that even in very simple settings GF does not maximize the margin even locally, and we believe that these results should be used as a starting point for studying which assumptions are required for proving margin maximization. Our positive results indeed show that under certain reasonable assumptions GF maximizes the margin (either locally or globally). Also, the notion of per-layer margin maximization which we consider suggests another path for obtaining positive results on the implicit bias.

In the paper, our focus is on understanding what can be guaranteed for the KKT convergence points specified in Lyu and Li [2019]. Accordingly, in most of our negative results, the construction assumes some specific initialization of GF, and does not quantify how "likely" they are to be reached under some random initialization. An exception is our negative result for depth-2 fully-connected ReLU networks (Theorem 3.2), which holds with constant probability under reasonable random initializations. Understanding whether this can be extended to the other settings we consider is an interesting problem for future research.

Our paper is structured as follows: In Section 2 we provide necessary notations and definitions, and discuss relevant prior results. Additional related works are discussed in Appendix A. In Sections 3, 4 and 5 we state our results on fully-connected networks, depth-2 networks in $\mathcal{N}$ and deep networks in $\mathcal{N}$ respectively, and provide some proof ideas. All formal proofs are deferred to Appendix C.

## 2 Preliminaries

**Notations.** We use bold-faced letters to denote vectors, e.g., $\mathbf{x} = (x_1, \ldots, x_d)$. For $\mathbf{x} \in \mathbb{R}^d$ we denote by $\|\mathbf{x}\|$ the Euclidean norm. We denote by $\mathbb{1}(\cdot)$ the indicator function, for example $\mathbb{1}(t \geq 5)$ equals 1 if $t \geq 5$ and 0 otherwise. For an integer $d \geq 1$ we denote $[d] = \{1, \ldots, d\}$.

**Neural networks.** A *fully-connected neural network* $\Phi$ of depth $m \geq 2$ is parameterized by a collection $\boldsymbol{\theta} = [W^{(l)}]_{l=1}^m$ of weight matrices, such that for every layer $l \in [m]$ we have $W^{(l)} \in \mathbb{R}^{d_l \times d_{l-1}}$. Thus, $d_l$ denotes the number of neurons in the $l$-th layer (i.e., the *width* of the layer). We

assume that $d_m = 1$ and denote by $d := d_0$ the input dimension. The neurons in layers $[m-1]$ are called *hidden neurons*. A fully-connected network computes a function $\Phi(\boldsymbol{\theta}; \cdot) : \mathbb{R}^d \to \mathbb{R}$ defined recursively as follows. For an input $\mathbf{x} \in \mathbb{R}^d$ we set $\mathbf{h}'_0 = \mathbf{x}$, and define for every $j \in [m-1]$ the input to the $j$-th layer as $\mathbf{h}_j = W^{(j)}\mathbf{h}'_{j-1}$, and the output of the $j$-th layer as $\mathbf{h}'_j = \sigma(\mathbf{h}_j)$, where $\sigma : \mathbb{R} \to \mathbb{R}$ is an activation function that acts coordinate-wise on vectors. Then, we define $\Phi(\boldsymbol{\theta}; \mathbf{x}) = W^{(m)}\mathbf{h}'_{m-1}$. Thus, there is no activation function in the output neuron. When considering depth-2 fully-connected networks we often use a parameterization $\boldsymbol{\theta} = [\mathbf{w}_1, \dots, \mathbf{w}_k, \mathbf{v}]$ where $\mathbf{w}_1, \dots, \mathbf{w}_k$ are the weights vectors of the $k$ hidden neurons (i.e., correspond to the rows of the first layer's weight matrix) and $\mathbf{v}$ are the weights of the second layer.

We also consider neural networks where some weights can be missing or shared. We define a class $\mathcal{N}$ of networks that may contain sparse and shared weights as follows. A network $\Phi$ in $\mathcal{N}$ is parameterized by $\boldsymbol{\theta} = [\mathbf{u}^{(l)}]_{l=1}^m$ where $m$ is the depth of $\Phi$, and $\mathbf{u}^{(l)} \in \mathbb{R}^{p_l}$ are the parameters of the $l$-th layer. We denote by $W^{(l)} \in \mathbb{R}^{d_l \times d_{l-1}}$ the weight matrix of the $l$-th layer. The matrix $W^{(l)}$ is described by the vector $\mathbf{u}^{(l)}$, and a function $g_l : [d_l] \times [d_{l-1}] \to [p_l] \cup \{0\}$ as follows: $W_{ij}^{(l)} = 0$ if $g_l(i, j) = 0$, and $W_{ij}^{(l)} = u_k$ if $g_l(i, j) = k > 0$. Thus, the function $g_l$ represents the sparsity and weight-sharing pattern of the $l$-th layer, and the dimension $p_l$ of $\mathbf{u}^{(l)}$ is the number of free parameters in the layer. We denote by $d := d_0$ the input dimension of the network and assume that the output dimension $d_m$ is 1. The function $\Phi(\boldsymbol{\theta}; \cdot) : \mathbb{R}^d \to \mathbb{R}$ computed by the neural network is defined recursively by the weight matrices as in the case of fully-connected networks. For example, convolutional neural networks are in $\mathcal{N}$. Note that the networks in $\mathcal{N}$ do not have bias terms and do not allow weight sharing between different layers. Moreover, we define a subclass $\mathcal{N}_{\text{no-share}}$ of $\mathcal{N}$, that contains networks without shared weights. Formally, a network $\Phi$ is in $\mathcal{N}_{\text{no-share}}$ if for every layer $l$ and every $k \in [p_l]$ there is at most one $(i, j) \in [d_l] \times [d_{l-1}]$ such that $g_l(i, j) = k$. Thus, networks in $\mathcal{N}_{\text{no-share}}$ might have sparse weights, but do not allow shared weights. For example, diagonal networks (defined below) and fully-connected networks are in $\mathcal{N}_{\text{no-share}}$.

A *diagonal neural network* is a network in $\mathcal{N}_{\text{no-share}}$ such that the weight matrix of each layer is diagonal, except for the last layer. Thus, the network is parameterized by $\boldsymbol{\theta} = [\mathbf{w}_1, \dots, \mathbf{w}_m]$ where $\mathbf{w}_j \in \mathbb{R}^d$ for all $j \in [m]$, and it computes a function $\Phi(\boldsymbol{\theta}; \cdot) : \mathbb{R}^d \to \mathbb{R}$ defined recursively as follows. For an input $\mathbf{x} \in \mathbb{R}^d$ set $\mathbf{h}_0 = \mathbf{x}$. For $j \in [m-1]$, the output of the $j$-th layer is $\mathbf{h}_j = \sigma(\text{diag}(\mathbf{w}_j)\mathbf{h}_{j-1})$. Then, we have $\Phi(\boldsymbol{\theta}; \mathbf{x}) = \mathbf{w}_m^\top \mathbf{h}_{m-1}$.

In all the above definitions the parameters $\boldsymbol{\theta}$ of the neural networks are given by a collection of matrices or vectors. We often view $\boldsymbol{\theta}$ as the vector obtained by concatenating the matrices or vectors in the collection. Thus, $\|\boldsymbol{\theta}\|$ denotes the $\ell_2$ norm of the vector $\boldsymbol{\theta}$.

The ReLU activation function is defined by $\sigma(z) = \max\{0, z\}$, and the linear activation is $\sigma(z) = z$. In this work we focus on ReLU networks (i.e., networks where all neurons have the ReLU activation) and on linear networks (where all neurons have the linear activation). We say that a network $\Phi$ is *homogeneous* if there exists $L > 0$ such that for every $\alpha > 0$ and $\boldsymbol{\theta}, \mathbf{x}$ we have $\Phi(\alpha\boldsymbol{\theta}; \mathbf{x}) = \alpha^L \Phi(\boldsymbol{\theta}; \mathbf{x})$. Note that in our definition of the class $\mathcal{N}$ we do not allow bias terms, and hence all linear and ReLU networks in $\mathcal{N}$ are homogeneous, where $L$ is the depth of the network. All networks considered in this work are homogeneous.

**Optimization problem and gradient flow (GF).** Let $S = \{(\mathbf{x}_i, y_i)\}_{i=1}^n \subseteq \mathbb{R}^d \times \{-1, 1\}$ be a binary classification training dataset. Let $\Phi$ be a neural network parameterized by $\boldsymbol{\theta} \in \mathbb{R}^m$. For a loss function $\ell : \mathbb{R} \to \mathbb{R}$ the empirical loss of $\Phi(\boldsymbol{\theta}; \cdot)$ on the dataset $S$ is

$$\mathcal{L}(\boldsymbol{\theta}) := \sum_{i=1}^n \ell(y_i \Phi(\boldsymbol{\theta}; \mathbf{x}_i)) . \tag{1}$$

We focus on the exponential loss $\ell(q) = e^{-q}$ and the logistic loss $\ell(q) = \log(1 + e^{-q})$.

We consider GF on the objective given in Eq. 1. This setting captures gradient descent with an infinitesimally small step size. Let $\boldsymbol{\theta}(t)$ be the trajectory of GF. Starting from an initial point $\boldsymbol{\theta}(0)$, the dynamics of $\boldsymbol{\theta}(t)$ is given by the differential equation $\frac{d\boldsymbol{\theta}(t)}{dt} = -\nabla \mathcal{L}(\boldsymbol{\theta}(t))$. Note that the ReLU function is not differentiable at 0. Practical implementations of gradient methods define the derivative $\sigma'(0)$ to be some constant in $[0, 1]$. We note that the exact value of $\sigma'(0)$ has no effect on our results.

**Convergence to a KKT point of the maximum-margin problem.** We say that a trajectory $\boldsymbol{\theta}(t)$ *converges in direction* to $\tilde{\boldsymbol{\theta}}$ if $\lim_{t\to\infty} \frac{\boldsymbol{\theta}(t)}{\|\boldsymbol{\theta}(t)\|} = \frac{\tilde{\boldsymbol{\theta}}}{\|\tilde{\boldsymbol{\theta}}\|}$. In this work we rely on the following theorem:

**Theorem 2.1** (Paraphrased from Lyu and Li [2019], Ji and Telgarsky [2020]). *Let $\Phi$ be a homogeneous linear or ReLU neural network. Consider minimizing either the exponential or the logistic loss over a binary classification dataset $\{(\mathbf{x}_i, y_i)\}_{i=1}^n$ using GF. Assume that there exists time $t_0$ such that $\mathcal{L}(\boldsymbol{\theta}(t_0)) < 1$, namely, $\Phi$ classifies every $\mathbf{x}_i$ correctly. Then, GF converges in direction to a first order stationary point (KKT point) of the following maximum margin problem in parameter space:*

$$\min_{\boldsymbol{\theta}} \frac{1}{2}\|\boldsymbol{\theta}\|^2 \quad s.t. \quad \forall i \in [n] \ \ y_i \Phi(\boldsymbol{\theta}; \mathbf{x}_i) \geq 1 \ . \tag{2}$$

*Moreover, $\mathcal{L}(\boldsymbol{\theta}(t)) \to 0$ and $\|\boldsymbol{\theta}(t)\| \to \infty$ as $t \to \infty$.*

In the case of ReLU networks, Problem 2 is non-smooth. Hence, the KKT conditions are defined using the Clarke subdifferential, which is a generalization of the derivative for non-differentiable functions. See Appendix B for a formal definition. We note that Lyu and Li [2019] proved the above theorem under the assumption that $\boldsymbol{\theta}$ converges in direction, and Ji and Telgarsky [2020] showed that such a directional convergence occurs and hence this assumption is not required.

Lyu and Li [2019] also showed that the KKT conditions of Problem 2 are necessary for optimality. In convex optimization problems, necessary KKT conditions are also sufficient for global optimality. However, the constraints in Problem 2 are highly non-convex. Moreover, the standard method for proving that necessary KKT conditions are sufficient for *local* optimality, is by showing that the KKT point satisfies certain *second order sufficient conditions (SOSC)* (cf. Ruszczynski [2011]). However, even when $\Phi$ is a linear neural network it is not known when such conditions hold. Thus, the KKT conditions of Problem 2 are not known to be sufficient even for local optimality.

A linear network with weight matrices $W^{(1)}, \ldots, W^{(m)}$ computes a linear predictor $\mathbf{x} \mapsto \langle \boldsymbol{\beta}, \mathbf{x} \rangle$ where $\boldsymbol{\beta} = W^{(m)} \cdot \ldots \cdot W^{(1)}$. Some prior works studied the implicit bias of linear networks in *predictor space*. Namely, characterizing the vector $\boldsymbol{\beta}$ from the aforementioned linear predictor. Gunasekar et al. [2018b] studied the implications of margin maximization in *parameter space* on the implicit bias in predictor space. They showed that minimizing $\|\boldsymbol{\theta}\|$ (under the constraints in Problem 2) implies: (1) Minimizing $\|\boldsymbol{\beta}\|_2$ for fully-connected networks; (2) Minimizing $\|\boldsymbol{\beta}\|_{2/L}$ for depth-$L$ diagonal networks; (3) Minimizing $\|\hat{\boldsymbol{\beta}}\|_{2/L}$ for depth-$L$ convolutional networks with full-dimensional filters, where $\hat{\boldsymbol{\beta}}$ are the Fourier coefficients of $\boldsymbol{\beta}$. However, these implications may not hold if GF converges to a KKT point which is not a global optimum of Problem 2.

For some classes of linear networks, positive results were obtained directly in predictor space, without assuming convergence to a global optimum of Problem 2 in parameter space. Most notably, for fully-connected linear networks (of any depth), Ji and Telgarsky [2020] showed that under the assumptions of Theorem 2.1, GF maximizes the $\ell_2$ margin in predictor space. Note that margin maximization in predictor space does not necessarily imply margin maximization in parameter space. Moreover, some results on the implicit bias in predictor space of linear convolutional networks with full-dimensional convolutional filters are given in Gunasekar et al. [2018b]. However, the architecture and set of assumptions are different than what we focus on. See Appendix A for a discussion on additional related work.

## 3 Fully-connected networks

First, we show that fully-connected linear networks converge to a global optimum of Problem 2.

**Theorem 3.1.** *Let $m \geq 2$ and let $\Phi$ be a depth-$m$ fully-connected linear network parameterized by $\boldsymbol{\theta}$. Consider minimizing either the exponential or the logistic loss over a dataset $\{(\mathbf{x}_i, y_i)\}_{i=1}^n$ using GF. Assume that there exists time $t_0$ such that $\mathcal{L}(\boldsymbol{\theta}(t_0)) < 1$. Then, GF converges in direction to a global optimum of Problem 2.*

*Proof idea (for the complete proof see Appendix C.2).* Building on results from Ji and Telgarsky [2020] and Du et al. [2018], we show that GF converges in direction to a KKT point $\tilde{\boldsymbol{\theta}} = [\tilde{W}^{(1)}, \ldots, \tilde{W}^{(m)}]$ such that for every $l \in [m]$ we have $\tilde{W}^{(l)} = C \cdot \mathbf{v}_l \mathbf{v}_{l-1}^\top$, where $C > 0$ and

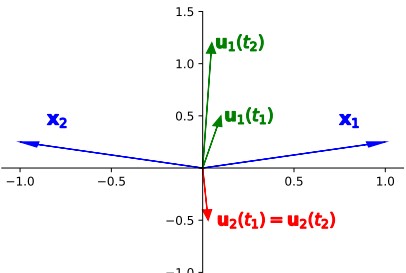

Figure 1: For time $t$ we denote $\mathbf{u}_i(t) = v_i(t)\mathbf{w}_i(t)$. Above we illustrate $\mathbf{u}_1(t), \mathbf{u}_2(t)$ for times $t_1 < t_2$. As $t \to \infty$ we have $\|\mathbf{u}_1(t)\| \to \infty$ and $\mathbf{u}_1$ converges in direction to $(0, 1)$. The vector $\mathbf{u}_2$ remains constant during the training. Hence $\frac{\mathbf{u}_2(t)}{\|\boldsymbol{\theta}(t)\|} \to \mathbf{0}$.

$\mathbf{v}_0, \ldots, \mathbf{v}_m$ are unit vectors (with $\mathbf{v}_m = 1$). Also, we have $\|\tilde{W}^{(m)} \cdot \ldots \cdot \tilde{W}^{(1)}\| = C^m = \min \|\mathbf{u}\|$ s.t. $y_i \mathbf{u}^\top \mathbf{x}_i \geq 1$ for all $i \in [n]$. Then, we show that every $\boldsymbol{\theta}$ that satisfies these properties, and satisfies the constraints of Problem 2, is a global optimum. Intuitively, the most "efficient" way (in terms of minimizing the parameters) to achieve margin 1 with a linear fully-connected network, is by using a network such that the direction of its corresponding linear predictor maximizes the margin, the layers are balanced (i.e., have equal norms), and the weight matrices of the layers are aligned. □

We now prove that the positive result in Theorem 3.1 does not apply to ReLU networks. We show that in depth-2 fully-connected ReLU networks GF might converge in direction to a KKT point of Problem 2 which is not even a local optimum. Moreover, it occurs under conditions holding with constant probability over reasonable random initializations.

**Theorem 3.2.** *Let $\Phi$ be a depth-2 fully-connected ReLU network with input dimension 2 and two hidden neurons. Namely, for $\boldsymbol{\theta} = [\mathbf{w}_1, \mathbf{w}_2, \mathbf{v}]$ and $\mathbf{x} \in \mathbb{R}^2$ we have $\Phi(\boldsymbol{\theta}; \mathbf{x}) = \sum_{l=1}^{2} v_l \sigma(\mathbf{w}_l^\top \mathbf{x})$. Consider minimizing either the exponential or the logistic loss using GF. Consider the dataset $\{(\mathbf{x}_1, y_1), (\mathbf{x}_2, y_2)\}$ where $\mathbf{x}_1 = \left(1, \frac{1}{4}\right)^\top$, $\mathbf{x}_2 = \left(-1, \frac{1}{4}\right)^\top$, and $y_1 = y_2 = 1$. Assume that the initialization $\boldsymbol{\theta}(0)$ is such that for every $i \in \{1, 2\}$ we have $\langle \mathbf{w}_1(0), \mathbf{x}_i \rangle > 0$ and $\langle \mathbf{w}_2(0), \mathbf{x}_i \rangle < 0$. Also, assume that $v_1(0) > 0$. Then, GF converges to zero loss, and converges in direction to a KKT point of Problem 2 which is not a local optimum.*

*Proof idea (for the complete proof see Appendix C.3).* By analyzing the dynamics of GF on the given dataset, we show that it converges to zero loss, and converges in direction to a KKT point $\tilde{\boldsymbol{\theta}}$ such that $\tilde{\mathbf{w}}_1 = (0, 2)^\top$, $\tilde{v}_1 = 2$, $\tilde{\mathbf{w}}_2 = \mathbf{0}$, and $\tilde{v}_2 = 0$. Note that $\tilde{\mathbf{w}}_2 = \mathbf{0}$ and $\tilde{v}_2 = 0$ since $\mathbf{w}_2(t), v_2(t)$ remain constant during the training and $\lim_{t \to \infty} \|\boldsymbol{\theta}(t)\| = \infty$. See Figure 1 for an illustration. Then, we show that for every $0 < \epsilon < 1$ there exists some $\boldsymbol{\theta}'$ such that $\|\boldsymbol{\theta}' - \tilde{\boldsymbol{\theta}}\| \leq \epsilon$, $\boldsymbol{\theta}'$ satisfies $y_i \Phi(\boldsymbol{\theta}'; \mathbf{x}_i) \geq 1$ for every $i \in \{1, 2\}$, and $\|\boldsymbol{\theta}'\| < \|\tilde{\boldsymbol{\theta}}\|$. Such $\boldsymbol{\theta}'$ is obtained from $\tilde{\boldsymbol{\theta}}$ by slightly changing $\tilde{\mathbf{w}}_1, \tilde{\mathbf{w}}_2$, and $\tilde{v}_2$. Thus, by using the second hidden neuron, which is not active in $\tilde{\boldsymbol{\theta}}$, we can obtain a solution $\boldsymbol{\theta}'$ with smaller norm. □

We note that the assumption on the initialization in the above theorem holds with constant probability for standard initialization schemes (e.g., Xavier initialization).

**Remark 3.1** (Unbounded sub-optimality)**.** *By choosing appropriate inputs $\mathbf{x}_1, \mathbf{x}_2$ in the setting of Theorem 3.2, it is not hard to show that the sub-optimality of the KKT point w.r.t. the global optimum can be arbitrarily large. Namely, for every large $M > 0$ we can choose a dataset where the angle between $\mathbf{x}_1$ and $\mathbf{x}_2$ is sufficiently close to $\pi$, such that $\frac{\|\tilde{\boldsymbol{\theta}}\|}{\|\boldsymbol{\theta}^*\|} \geq M$, where $\tilde{\boldsymbol{\theta}}$ is a KKT point to which GF converges, and $\boldsymbol{\theta}^*$ is a global optimum of Problem 2. Indeed, as illustrated in Figure 1, if one neuron is active on both inputs and the other neuron is not active on any input, then the active neuron needs to be very large in order to achieve margin 1, while if each neuron is active on a single input then we can achieve margin 1 with much smaller parameters. We note that such unbounded sub-optimality can be obtained also in other negative results in this work (in Theorems 4.1, 4.3, 4.4 and 5.4).*

**Remark 3.2** (Robustness to small perturbations)**.** *Theorem 3.2 holds even if we slightly perturb the inputs $\mathbf{x}_1, \mathbf{x}_2$. Thus, it is not sensitive to small changes in the dataset. We note that such robustness to small perturbations can be shown also for the negative results in Theorems 4.1, 4.3, 5.1 and 5.4.*

# 4 Depth-2 networks in $\mathcal{N}$

In this section we study depth-2 linear and ReLU networks in $\mathcal{N}$. We first show that already for linear networks in $\mathcal{N}_{\text{no-share}}$ (more specifically, for diagonal networks) GF may not converge even to a local optimum.

**Theorem 4.1.** *Let $\Phi$ be a depth-2 linear or ReLU diagonal neural network parameterized by $\boldsymbol{\theta} = [\mathbf{w}_1, \mathbf{w}_2]$. Consider minimizing either the exponential or the logistic loss using GF. There exists a dataset $\{(\mathbf{x}, y)\} \subseteq \mathbb{R}^2 \times \{-1, 1\}$ of size 1 and an initialization $\boldsymbol{\theta}(0)$, such that GF converges to zero loss, and converges in direction to a KKT point $\tilde{\boldsymbol{\theta}}$ of Problem 2 which is not a local optimum.*

*Proof idea (for the complete proof see Appendix C.4).* Let $\mathbf{x} = (1, 2)^\top$ and $y = 1$. Let $\boldsymbol{\theta}(0)$ such that $\mathbf{w}_1(0) = \mathbf{w}_2(0) = (1, 0)^\top$. Recalling that the diagonal network computes the function $\mathbf{x} \mapsto (\mathbf{w}_1 \circ \mathbf{w}_2)^\top \mathbf{x}$ (where $\circ$ is the entry-wise product), we see that the second coordinate remains inactive during training. It is not hard to show that GF converges to the KKT point $\tilde{\boldsymbol{\theta}}$ with $\tilde{\mathbf{w}}_1 = \tilde{\mathbf{w}}_2 = (1, 0)^\top$. However, it is not a local optimum, since for small $\epsilon > 0$ the parameters $\boldsymbol{\theta}' = [\mathbf{w}_1', \mathbf{w}_2']$ with $\mathbf{w}_1' = \mathbf{w}_2' = \left(\sqrt{1 - \epsilon}, \sqrt{\frac{\epsilon}{2}}\right)^\top$ satisfy the constraints of Problem 2, and we have $\|\boldsymbol{\theta}'\| < \|\tilde{\boldsymbol{\theta}}\|$. $\qquad\square$

By Theorem 3.2 fully-connected ReLU networks may not converge to a local optimum, and by Theorem 4.1 linear (and ReLU) networks with sparse weights may not converge to a local optimum. In the proofs of both of these negative results, GF converges to a KKT point such that one of the weights vectors of the hidden neurons is zero. However, in practice gradient descent often converges to a network that does not contain such disconnected neurons. Hence, a natural question is whether the negative results hold also in networks that do not contain neurons whose weights vector is zero. In the following theorem we show that in linear networks such an assumption allows us to obtain a positive result. Namely, in depth-2 linear networks in $\mathcal{N}_{\text{no-share}}$, if GF converges to a KKT point of Problem 2 that satisfies this condition, then it is guaranteed to be a global optimum. However, we also show that in ReLU networks assuming that all neurons have non-zero weights is not sufficient.

**Theorem 4.2.** *We have:*

1. *Let $\Phi$ be a depth-2 linear neural network in $\mathcal{N}_{\text{no-share}}$ parameterized by $\boldsymbol{\theta}$. Consider minimizing either the exponential or the logistic loss over a dataset $\{(\mathbf{x}_i, y_i)\}_{i=1}^n$ using GF. Assume that there exists time $t_0$ such that $\mathcal{L}(\boldsymbol{\theta}(t_0)) < 1$, and let $\tilde{\boldsymbol{\theta}}$ be the KKT point of Problem 2 such that $\boldsymbol{\theta}(t)$ converges to $\tilde{\boldsymbol{\theta}}$ in direction (such $\tilde{\boldsymbol{\theta}}$ exists by Theorem 2.1). Assume that in the network parameterized by $\tilde{\boldsymbol{\theta}}$ all hidden neurons have non-zero incoming weights vectors. Then, $\tilde{\boldsymbol{\theta}}$ is a global optimum of Problem 2.*

2. *Let $\Phi$ be a fully-connected depth-2 ReLU network with input dimension 2 and 4 hidden neurons parameterized by $\boldsymbol{\theta}$. Consider minimizing either the exponential or the logistic loss using GF. There exists a dataset and an initialization $\boldsymbol{\theta}(0)$, such that GF converges to zero loss, and converges in direction to a KKT point $\tilde{\boldsymbol{\theta}}$ of Problem 2, which is not a local optimum, and in the network parameterized by $\tilde{\boldsymbol{\theta}}$ all hidden neurons have non-zero incoming weights.*

*Proof idea (for the complete proof see Appendix C.5).* We give here the proof idea for part (1). Let $k$ be the width of the network. For every $j \in [k]$ we denote by $\mathbf{w}_j$ the incoming weights vector to the $j$-th hidden neuron, and by $v_j$ the outgoing weight. Let $\mathbf{u}_j = v_j \mathbf{w}_j$. We consider an optimization problem over the variables $\mathbf{u}_1, \ldots, \mathbf{u}_k$ where the objective is to minimize $\sum_{j \in [k]} \|\mathbf{u}_j\|$ and the constrains correspond to the constraints of Problem 2. Let $\tilde{\boldsymbol{\theta}} = [\tilde{\mathbf{w}}_1, \ldots, \tilde{\mathbf{w}}_k, \tilde{\mathbf{v}}]$ be the KKT point of Problem 2 to which GF converges in direction. For every $j \in [k]$ we denote $\tilde{\mathbf{u}}_j = \tilde{v}_j \tilde{\mathbf{w}}_j$. We show that $\tilde{\mathbf{u}}_1, \ldots, \tilde{\mathbf{u}}_k$ satisfy the KKT conditions of the aforementioned problem. Since the objective there is convex and the constrains are affine, then it is a global optimum. Finally, we show that it implies global optimality of $\tilde{\boldsymbol{\theta}}$. $\qquad\square$

**Remark 4.1** (Implications on margin maximization in predictor space for diagonal linear networks)**.** *Theorems 4.1 and 4.2 imply analogous results on diagonal linear networks also in predictor space. As we discussed in Section 2, Gunasekar et al. [2018b] showed that in depth-2 diagonal linear networks, minimizing $\|\boldsymbol{\theta}\|_2$ under the constraints in Problem 2 implies minimizing $\|\boldsymbol{\beta}\|_1$, where $\boldsymbol{\beta}$ is the corresponding linear predictor. Theorem 4.1 can be easily extended to predictor space, namely,*

*GF on depth-2 linear diagonal networks might converge to a KKT point $\tilde{\boldsymbol{\theta}}$ of Problem 2, such that the corresponding linear predictor $\tilde{\boldsymbol{\beta}}$ is not an optimum of the following problem:*

$$\underset{\boldsymbol{\beta}}{\operatorname{argmin}} \|\boldsymbol{\beta}\|_1 \quad s.t. \quad \forall i \in [n] \ y_i \langle \boldsymbol{\beta}, \mathbf{x}_i \rangle \geq 1 . \tag{3}$$

*Moreover, by combining part (1) of Theorem 4.2 with the result from Gunasekar et al. [2018b], we deduce that if GF on a depth-2 diagonal linear network converges to a KKT point $\tilde{\boldsymbol{\theta}}$ of Problem 2 with non-zero weights vectors, then the corresponding linear predictor is a global optimum of Problem 3.*

We argue that since in practice gradient descent often converges to networks without zero-weight neurons, then part (1) of Theorem 4.2 gives a useful positive result for depth-2 linear networks. However, by part (2) of Theorem 4.2, this assumption is not sufficient for obtaining a positive result in the case of ReLU networks. Hence, we now consider a stronger assumption, namely, that the KKT point $\tilde{\boldsymbol{\theta}}$ is such that for every $\mathbf{x}_i$ in the dataset the inputs to all hidden neurons in the computation $\Phi(\tilde{\boldsymbol{\theta}}; \mathbf{x}_i)$ are non-zero. In the following theorem we show that in depth-2 ReLU networks, if the KKT point satisfies this condition then it is guaranteed to be a local optimum of Problem 2. However, even under this condition it is not necessarily a global optimum. The proof is given in Appendix C.6 and uses ideas from the previous proofs, with some required modifications.

**Theorem 4.3.** *Let $\Phi$ be a depth-2 ReLU network in $\mathcal{N}_{no\text{-}share}$ parameterized by $\boldsymbol{\theta}$. Consider minimizing either the exponential or the logistic loss over a dataset $\{(\mathbf{x}_i, y_i)\}_{i=1}^n$ using GF. Assume that there exists time $t_0$ such that $\mathcal{L}(\boldsymbol{\theta}(t_0)) < 1$, and let $\tilde{\boldsymbol{\theta}}$ be the KKT point of Problem 2 such that $\boldsymbol{\theta}(t)$ converges to $\tilde{\boldsymbol{\theta}}$ in direction (such $\tilde{\boldsymbol{\theta}}$ exists by Theorem 2.1). Assume that for every $i \in [n]$ the inputs to all hidden neurons in the computation $\Phi(\tilde{\boldsymbol{\theta}}; \mathbf{x}_i)$ are non-zero. Then, $\tilde{\boldsymbol{\theta}}$ is a local optimum of Problem 2. However, it may not be a global optimum, even if the network $\Phi$ is fully connected.*

Note that in all the above theorems we do not allow shared weights. We now consider the case of depth-2 linear or ReLU networks in $\mathcal{N}$, where the first layer is convolutional with disjoint patches (and hence has shared weights), and show that GF does not always converge in direction to a local optimum, even when the inputs to all hidden neurons are non-zero (and hence there are no zero weights vectors).

**Theorem 4.4.** *Let $\Phi$ be a depth-2 linear or ReLU network in $\mathcal{N}$, parameterized by $\boldsymbol{\theta} = [\mathbf{w}, \mathbf{v}]$ for $\mathbf{w}, \mathbf{v} \in \mathbb{R}^2$, such that for $\mathbf{x} \in \mathbb{R}^4$ we have $\Phi(\boldsymbol{\theta}; \mathbf{x}) = \sum_{j=1}^2 v_j \sigma(\mathbf{w}^\top \mathbf{x}^{(j)})$ where $\mathbf{x}^{(1)} = (x_1, x_2)$ and $\mathbf{x}^{(2)} = (x_3, x_4)$. Thus, $\Phi$ is a convolutional network with two disjoint patches. Consider minimizing either the exponential or the logistic loss using GF. Then, there exists a dataset $\{(\mathbf{x}, y)\}$ of size 1, and an initialization $\boldsymbol{\theta}(0)$, such that GF converges to zero loss, and converges in direction to a KKT point $\tilde{\boldsymbol{\theta}} = [\tilde{\mathbf{w}}, \tilde{\mathbf{v}}]$ of Problem 2 which is not a local optimum. Moreover, $\langle \tilde{\mathbf{w}}, \mathbf{x}^{(j)} \rangle \neq 0$ for $j \in \{1, 2\}$.*

*Proof idea (for the complete proof see Appendix C.7). Let $\mathbf{x} = \left( 4, \frac{1}{\sqrt{2}}, -4, \frac{1}{\sqrt{2}} \right)^\top$ and $y = 1$. Let $\boldsymbol{\theta}(0)$ such that $\mathbf{w}(0) = (0, 1)^\top$ and $\mathbf{v}(0) = \left( \frac{1}{\sqrt{2}}, \frac{1}{\sqrt{2}} \right)^\top$. Since $\mathbf{x}^{(1)}$ and $\mathbf{x}^{(2)}$ are symmetric w.r.t. $\mathbf{w}(0)$, and $\mathbf{v}(0)$ does not break this symmetry, then $\mathbf{w}$ keeps its direction throughout the training. Thus, we show that GF converges in direction to a KKT point $\tilde{\boldsymbol{\theta}}$ where $\tilde{\mathbf{w}} = (0, 1)^\top$ and $\tilde{\mathbf{v}} = \left( \frac{1}{\sqrt{2}}, \frac{1}{\sqrt{2}} \right)^\top$. Then, we show that it is not a local optimum, since for every small $\epsilon > 0$ the parameters $\boldsymbol{\theta}' = [\mathbf{w}', \mathbf{v}']$ with $\mathbf{w}' = (\sqrt{\epsilon}, 1 - \epsilon)^\top$ and $\mathbf{v}' = \left( \frac{1}{\sqrt{2}} + \frac{\sqrt{\epsilon}}{2}, \frac{1}{\sqrt{2}} - \frac{\sqrt{\epsilon}}{2} \right)^\top$ satisfy the constraints of Problem 2, and we have $\|\boldsymbol{\theta}'\| < \|\tilde{\boldsymbol{\theta}}\|$.* $\qquad \square$

## 5 Deep networks in $\mathcal{N}$

In this section we study the more general case of depth-$m$ neural networks in $\mathcal{N}$, where $m \geq 2$. First, we show that for networks of depth at least 3 in $\mathcal{N}_{no\text{-}share}$, GF may not converge to a local optimum of Problem 2, for both linear and ReLU networks, and even where there are no zero weights vectors and the inputs to all hidden neurons are non-zero. More precisely, we prove this claim for diagonal networks.

**Theorem 5.1.** *Let $m \geq 3$. Let $\Phi$ be a depth-$m$ linear or ReLU diagonal neural network parameterized by $\boldsymbol{\theta}$. Consider minimizing either the exponential or the logistic loss using GF. There exists a dataset $\{(\mathbf{x}, y)\} \subseteq \mathbb{R}^2 \times \{-1, 1\}$ of size 1 and an initialization $\boldsymbol{\theta}(0)$, such that GF converges to zero loss, and converges in direction to a KKT point $\tilde{\boldsymbol{\theta}}$ of Problem 2 which is not a local optimum. Moreover, all inputs to neurons in the computation $\Phi(\tilde{\boldsymbol{\theta}}; \mathbf{x})$ are non-zero.*

*Proof idea (for the complete proof see Appendix C.8).* Let $\mathbf{x} = (1, 1)^\top$ and $y = 1$. Consider the initialization $\boldsymbol{\theta}(0)$ where $\mathbf{w}_j(0) = (1, 1)^\top$ for every $j \in [m]$. We show that GF converges in direction to a KKT point $\tilde{\boldsymbol{\theta}} = [\tilde{\mathbf{w}}_1, \ldots, \tilde{\mathbf{w}}_m]$ such that $\tilde{\mathbf{w}}_j = \left(2^{-1/m}, 2^{-1/m}\right)^\top$ for all $j \in [m]$. Then, we consider the parameters $\boldsymbol{\theta}' = [\mathbf{w}_1', \ldots, \mathbf{w}_m']$ such that for every $j \in [m]$ we have $\mathbf{w}_j' = \left(\left(\frac{1+\epsilon}{2}\right)^{1/m}, \left(\frac{1-\epsilon}{2}\right)^{1/m}\right)^\top$, and show that if $\epsilon > 0$ is sufficiently small, then $\boldsymbol{\theta}'$ satisfies the constraints in Problem 2 and we have $\|\boldsymbol{\theta}'\| < \|\tilde{\boldsymbol{\theta}}\|$. $\qquad\square$

Note that in the case of linear networks, the above result is in contrast to networks with sparse weights of depth 2 that converge to a global optimum by Theorem 4.2, and to fully-connected networks of any depth that converge to a global optimum by Theorem 3.1. In the case of ReLU networks, the above result is in contrast to the case of depth-2 networks studied in Theorem 4.3, where it is guaranteed to converge to a local optimum.

In light of our negative results, we now consider a weaker notion of margin maximization, namely, maximizing the margin for each layer separately. Let $\Phi$ be a neural network of depth $m$ in $\mathcal{N}$, parameterized by $\boldsymbol{\theta} = [\mathbf{u}^{(l)}]_{l=1}^m$. The maximum margin problem for a layer $l_0 \in [m]$ w.r.t. $\boldsymbol{\theta}_0 = [\mathbf{u}_0^{(l)}]_{l=1}^m$ is the following:

$$\min_{\mathbf{u}^{(l_0)}} \frac{1}{2} \left\| \mathbf{u}^{(l_0)} \right\|^2 \quad \text{s.t.} \quad \forall i \in [n] \ y_i \Phi(\boldsymbol{\theta}'; \mathbf{x}_i) \geq 1 , \tag{4}$$

where $\boldsymbol{\theta}' = [\mathbf{u}_0^{(1)}, \ldots, \mathbf{u}_0^{(l_0-1)}, \mathbf{u}^{(l_0)}, \mathbf{u}_0^{(l_0+1)}, \ldots, \mathbf{u}_0^{(m)}]$. For linear networks we have the following:

**Theorem 5.2.** *Let $m \geq 2$. Let $\Phi$ be any depth-$m$ linear neural network in $\mathcal{N}$, parameterized by $\boldsymbol{\theta} = [\mathbf{u}^{(l)}]_{l=1}^m$. Consider minimizing either the exponential or the logistic loss over a dataset $\{(\mathbf{x}_i, y_i)\}_{i=1}^n$ using GF. Assume that there exists time $t_0$ such that $\mathcal{L}(\boldsymbol{\theta}(t_0)) < 1$. Then, GF converges in direction to a KKT point $\tilde{\boldsymbol{\theta}} = [\tilde{\mathbf{u}}^{(l)}]_{l=1}^m$ of Problem 2, such that for every layer $l \in [m]$ the parameters vector $\tilde{\mathbf{u}}^{(l)}$ is a global optimum of Problem 4 w.r.t. $\tilde{\boldsymbol{\theta}}$.*

The theorem follows by noticing that if $\Phi$ is a linear network, then the constraints in Problem 4 are affine, and its KKT conditions are implied by the KKT conditions of Problem 2. See Appendix C.9 for the formal proof. Note that by Theorems 4.1, 4.4 and 5.1, linear networks in $\mathcal{N}$ might converge in direction to a KKT point $\tilde{\boldsymbol{\theta}}$, which is not a local optimum of Problem 2. However, Theorem 5.2 implies that each layer in $\tilde{\boldsymbol{\theta}}$ is a global optimum of Problem 4. Hence, any improvement to $\tilde{\boldsymbol{\theta}}$ requires changing at least two layers simultaneously.

While in linear networks GF maximize the margin for each layer separately, in the following theorem (which we prove in Appendix C.10) we show that this claim does not hold for ReLU networks: Already for fully-connected networks of depth-2 GF may not converge in direction to a local optimum of Problem 4.

**Theorem 5.3.** *Let $\Phi$ be a fully-connected depth-2 ReLU network with input dimension 2 and 4 hidden neurons parameterized by $\boldsymbol{\theta}$. Consider minimizing either the exponential or the logistic loss using GF. There exists a dataset and an initialization $\boldsymbol{\theta}(0)$ such that GF converges to zero loss, and converges in direction to a KKT point $\tilde{\boldsymbol{\theta}}$ of Problem 2, such that the weights of the first layer are not a local optimum of Problem 4 w.r.t. $\tilde{\boldsymbol{\theta}}$.*

Finally, we show that in ReLU networks in $\mathcal{N}$ of any depth, if the KKT point to which GF converges in direction is such that the inputs to hidden neurons are non-zero, then it must be a local optimum of Problem 4 (but not necessarily a global optimum). The proof follows the ideas from the proof of Theorem 5.2, with some required modifications, and is given in Appendix C.11.

**Theorem 5.4.** *Let $m \geq 2$. Let $\Phi$ be any depth-$m$ ReLU network in $\mathcal{N}$ parameterized by $\boldsymbol{\theta} = [\mathbf{u}^{(l)}]_{l=1}^m$. Consider minimizing either the exponential or the logistic loss over a dataset $\{(\mathbf{x}_i, y_i)\}_{i=1}^n$ using GF, and assume that there exists time $t_0$ such that $\mathcal{L}(\boldsymbol{\theta}(t_0)) < 1$. Let $\tilde{\boldsymbol{\theta}} = [\tilde{\mathbf{u}}^{(l)}]_{l=1}^m$ be the KKT point of Problem 2 such that $\boldsymbol{\theta}(t)$ converges to $\tilde{\boldsymbol{\theta}}$ in direction (such $\tilde{\boldsymbol{\theta}}$ exists by Theorem 2.1). Let $l \in [m]$ and assume that for every $i \in [n]$ the inputs to all neurons in layers $\geq l$ in the computation $\Phi(\tilde{\boldsymbol{\theta}}; \mathbf{x}_i)$ are non-zero. Then, the parameters vector $\tilde{\mathbf{u}}^{(l)}$ is a local optimum of Problem 4 w.r.t. $\tilde{\boldsymbol{\theta}}$. However, it may not be a global optimum.*

## Acknowledgments and Disclosure of Funding

This research is supported in part by European Research Council (ERC) grant 754705.

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
