# A   Additional related work

Soudry et al. [2018] showed that gradient descent on linearly-separable binary classification problems with exponentially-tailed losses (e.g., the exponential loss and the logistic loss), converges to the maximum $\ell_2$-margin direction. This analysis was extended to other loss functions, tighter convergence rates, non-separable data, and variants of gradient-based optimization algorithms [Nacson et al., 2019, Ji and Telgarsky, 2018b, Ji et al., 2020, Gunasekar et al., 2018a, Shamir, 2020, Ji and Telgarsky, 2021].

As detailed in Section 2, Lyu and Li [2019] and Ji and Telgarsky [2020] showed that GF on homogeneous neural networks with exponential-type losses converge in direction to a KKT point of the maximum-margin problem in parameter space. The implications of margin maximization in parameter space on the implicit bias in predictor space for linear neural networks were studied in Gunasekar et al. [2018b] (as detailed in Section 2) and also in Jagadeesan et al. [2021], Ergen and Pilanci [2021a,b]. Moreover, several recent works considered implications of convergence to a KKT point of the maximum-margin problem, without assuming that the KKT point is optimal: Safran et al. [2022] proved a generalization bound in univariate depth-2 ReLU networks, Vardi et al. [2022] proved bias towards non-robust solutions in depth-2 ReLU networks, and Haim et al. [2022] showed that training data can be reconstructed from trained networks. Margin maximization in predictor space for fully-connected linear networks was shown by Ji and Telgarsky [2020] (as detailed in Section 2), and similar results under stronger assumptions were previously established in Gunasekar et al. [2018b] and in Ji and Telgarsky [2018a]. The implicit bias in predictor space of diagonal and convolutional linear networks was studied in Gunasekar et al. [2018b], Moroshko et al. [2020], Yun et al. [2020]. Chizat and Bach [2020] studied the dynamics of GF on infinite-width homogeneous two-layer networks with exponentially-tailed losses, and showed bias towards margin maximization w.r.t. a certain function norm known as the variation norm. Sarussi et al. [2021] studied GF on two-layer leaky-ReLU networks, where the training data is linearly separable, and showed convergence to a linear classifier based on a certain assumption called *Neural Agreement Regime (NAR)*. Phuong and Lampert [2020] studied the implicit bias in depth-2 ReLU networks trained on *orthogonally-separable* data.

Lyu et al. [2021] studied the implicit bias in two-layer leaky-ReLU networks trained on linearly separable and symmetric data, and showed that GF converges to a linear classifier which maximizes the $\ell_2$ margin. They also gave constructions where a KKT point is not a global max-margin solution. We note that their constructions do not imply any of our results. In particular, for the ReLU activation they showed a construction where there exists a KKT point which is not a global optimum of the max-margin problem, however this KKT point is not reachable with GF. Thus, there does not exist an initialization such that GF actually converges to this point. In our construction (Theorem 3.2) GF converges to a suboptimal KKT point with constant probability over the initialization. Moreover, even their construction for the leaky-ReLU activation (which is their main focus) considers only global suboptimality, while we show local suboptimality for the ReLU activation.

Finally, the implicit bias of neural networks in regression tasks w.r.t. the square loss was also extensively studied in recent years (e.g., Gunasekar et al. [2018c], Razin and Cohen [2020], Arora et al. [2019], Belabbas [2020], Eftekhari and Zygalakis [2020], Li et al. [2018], Ma et al. [2018], Woodworth et al. [2020], Gidel et al. [2019], Li et al. [2020], Yun et al. [2020], Vardi and Shamir [2021], Azulay et al. [2021], Timor et al. [2022]). This setting, however, is less relevant to our work.

For a broader discussion on the implicit bias in neural networks, in both classification and regression tasks, see a survey in Vardi [2022].

# B   Preliminaries on the KKT conditions

Below we review the definition of the KKT condition for non-smooth optimization problems (cf. Lyu and Li [2019], Dutta et al. [2013]).

Let $f : \mathbb{R}^d \to \mathbb{R}$ be a locally Lipschitz function. The Clarke subdifferential [Clarke et al., 2008] at $\mathbf{x} \in \mathbb{R}^d$ is the convex set

$$\partial^\circ f(\mathbf{x}) := \text{conv} \left\{ \lim_{i \to \infty} \nabla f(\mathbf{x}_i) \;\middle|\; \lim_{i \to \infty} \mathbf{x}_i = \mathbf{x}, \; f \text{ is differentiable at } \mathbf{x}_i \right\} .$$

If $f$ is continuously differentiable at $\mathbf{x}$ then $\partial^\circ f(\mathbf{x}) = \{\nabla f(\mathbf{x})\}$.

Consider the following optimization problem

$$\min f(\mathbf{x}) \quad \text{s.t.} \quad \forall n \in [N] \;\; g_n(\mathbf{x}) \leq 0 \,, \tag{5}$$

where $f, g_1, \ldots, g_n : \mathbb{R}^d \to \mathbb{R}$ are locally Lipschitz functions. We say that $\mathbf{x} \in \mathbb{R}^d$ is a feasible point of Problem 5 if $\mathbf{x}$ satisfies $g_n(\mathbf{x}) \leq 0$ for all $n \in [N]$. We say that a feasible point $\mathbf{x}$ is a KKT point if there exists $\lambda_1, \ldots, \lambda_N \geq 0$ such that

1. $\mathbf{0} \in \partial^\circ f(\mathbf{x}) + \sum_{n \in [N]} \lambda_n \partial^\circ g_n(\mathbf{x})$;

2. For all $n \in [N]$ we have $\lambda_n g_n(\mathbf{x}) = 0$.

## C   Proofs

### C.1   Auxiliary lemmas

Throughout our proofs we use the following two lemmas from Du et al. [2018]:

**Lemma C.1** (Du et al. [2018])**.** *Let $m \geq 2$, and let $\Phi$ be a depth-$m$ fully-connected linear or ReLU network parameterized by $\boldsymbol{\theta} = [W_1, \ldots, W_m]$. Suppose that for every $j \in [m]$ we have $W_j \in \mathbb{R}^{d_j \times d_{j-1}}$. Consider minimizing any differentiable loss function (e.g., the exponential or the logistic loss) over a dataset using GF. Then, for every $j \in [m-1]$ at all time $t$ we have*

$$\frac{d}{dt}\left(\|W_j\|_F^2 - \|W_{j+1}\|_F^2\right) = 0 \,.$$

*Moreover, for every $j \in [m-1]$ and $i \in [d_j]$ we have*

$$\frac{d}{dt}\left(\|W_j[i,:]\|^2 - \|W_{j+1}[:,i]\|^2\right) = 0 \,,$$

*where $W_j[i,:]$ is the vector of incoming weights to the $i$-th neuron in the $j$-th hidden layer (i.e., the $i$-th row of $W_j$), and $W_{j+1}[:,i]$ is the vector of outgoing weights from this neuron (i.e., the $i$-th column of $W_{j+1}$).*

**Lemma C.2** (Du et al. [2018])**.** *Let $m \geq 2$, and let $\Phi$ be a depth-$m$ linear or ReLU network in $\mathcal{N}$, parameterized by $\boldsymbol{\theta} = [\mathbf{u}^{(1)}, \ldots, \mathbf{u}^{(m)}]$. Consider minimizing any differentiable loss function (e.g., the exponential or the logistic loss) over a dataset using GF. Then, for every $j \in [m-1]$ at all time $t$ we have*

$$\frac{d}{dt}\left(\left\|\mathbf{u}^{(j)}\right\|^2 - \left\|\mathbf{u}^{(j+1)}\right\|^2\right) = 0 \,.$$

Note that Lemma C.2 considers a larger family of neural networks since it allows sparse and shared weights, but Lemma C.1 gives a stronger guarantee, since it implies balancedness between the incoming and outgoing weights of each hidden neuron separately. In our proofs we will also need to use a balancedness property for each hidden neuron separately in depth-2 networks with sparse weights. Since this property is not implied by the above lemmas from Du et al. [2018], we now prove it.

Before stating the lemma, let us introduce some required notations. Let $\Phi$ be a depth-2 network in $\mathcal{N}_{\text{no-share}}$. We can always assume w.l.o.g. that the second layer is fully connected, namely, all hidden neurons are connected to the output neuron. Indeed, otherwise we can ignore the neurons that are not connected to the output neuron. For the network $\Phi$ we use the parameterization $\boldsymbol{\theta} = [\mathbf{w}_1, \ldots, \mathbf{w}_k, \mathbf{v}]$, where $k$ is the number of hidden neurons. For every $j \in [k]$ the vector $\mathbf{w}_j \in \mathbb{R}^{p_j}$ is the weights vector of the $j$-th hidden neuron, and we have $1 \leq p_j \leq d$ where $d$ is the input dimention. For an input $\mathbf{x} \in \mathbb{R}^d$ we denote by $\mathbf{x}^j \in \mathbb{R}^{p_j}$ a sub-vector of $\mathbf{x}$, such that $\mathbf{x}^j$ includes the coordinates of $\mathbf{x}$ that are connected to the $j$-th hidden neuron. Thus, given $\mathbf{x}$, the input to the $j$-th hidden neuron is $\langle \mathbf{w}_j, \mathbf{x}^j \rangle$. The vector $\mathbf{v} \in \mathbb{R}^k$ is the weights vector of the second layer. Overall, we have $\Phi(\boldsymbol{\theta}; \mathbf{x}) = \sum_{j \in [k]} v_j \sigma(\mathbf{w}_j^\top \mathbf{x}^j)$.

**Lemma C.3.** *Let $\Phi$ be a depth-2 linear or ReLU network in $\mathcal{N}_{\text{no-share}}$, parameterized by $\boldsymbol{\theta} = [\mathbf{w}_1, \ldots, \mathbf{w}_k, \mathbf{v}]$. Consider minimizing any differentiable loss function (e.g., the exponential or the logistic loss) over a dataset using GF. Then, for every $j \in [k]$ at all time $t$ we have*

$$\frac{d}{dt}\left(\|\mathbf{w}_j\|^2 - v_j^2\right) = 0 \,.$$

*Proof.* We have

$$\mathcal{L}(\boldsymbol{\theta}) = \sum_{i\in[n]} \ell\left(y_i \Phi(\boldsymbol{\theta}; \mathbf{x}_i)\right) = \sum_{i\in[n]} \ell\left(y_i \sum_{l\in[k]} v_l \sigma(\mathbf{w}_l^\top \mathbf{x}_i^j)\right).$$

Hence

$$\frac{d}{dt}\left(\|\mathbf{w}_j\|^2\right) = 2\langle \mathbf{w}_j, \frac{d\mathbf{w}_j}{dt}\rangle = -2\langle \mathbf{w}_j, \nabla_{\mathbf{w}_j}\mathcal{L}(\boldsymbol{\theta})\rangle$$

$$= -2\sum_{i\in[n]} \ell'\left(y_i \sum_{l\in[k]} v_l \sigma(\mathbf{w}_l^\top \mathbf{x}_i^l)\right) \cdot y_i v_j \sigma'(\mathbf{w}_j^\top \mathbf{x}_i^j)\mathbf{w}_j^\top \mathbf{x}_i^j$$

$$= -2\sum_{i\in[n]} \ell'\left(y_i \sum_{l\in[k]} v_l \sigma(\mathbf{w}_l^\top \mathbf{x}_i^l)\right) \cdot y_i v_j \sigma(\mathbf{w}_j^\top \mathbf{x}_i^j).$$

Moreover,

$$\frac{d}{dt}\left(v_j^2\right) = 2v_j \frac{dv_j}{dt} = -2v_j \nabla_{v_j}\mathcal{L}(\boldsymbol{\theta})$$

$$= -2v_j \sum_{i\in[n]} \ell'\left(y_i \sum_{l\in[k]} v_l \sigma(\mathbf{w}_l^\top \mathbf{x}_i^l)\right) \cdot y_i \sigma(\mathbf{w}_j^\top \mathbf{x}_i^j).$$

Hence the lemma follows. $\qquad\square$

Using the above lemma, we show the following:

**Lemma C.4.** *Let $\Phi$ be a depth-2 linear or ReLU network in $\mathcal{N}_{no\text{-}share}$, parameterized by $\boldsymbol{\theta} = [\mathbf{w}_1, \ldots, \mathbf{w}_k, \mathbf{v}]$. Consider minimizing any differentiable loss function (e.g., the exponential or the logistic loss) over a dataset using GF starting from $\boldsymbol{\theta}(0)$. Assume that $\lim_{t\to\infty}\|\boldsymbol{\theta}(t)\| = \infty$ and that $\boldsymbol{\theta}(t)$ converges in direction to $\tilde{\boldsymbol{\theta}} = [\tilde{\mathbf{w}}_1, \ldots, \tilde{\mathbf{w}}_k, \tilde{\mathbf{v}}]$, i.e., $\tilde{\boldsymbol{\theta}} = \|\tilde{\boldsymbol{\theta}}\| \cdot \lim_{t\to\infty}\frac{\boldsymbol{\theta}(t)}{\|\boldsymbol{\theta}(t)\|}$. Then, for every $l \in [k]$ we have $\|\tilde{\mathbf{w}}_l\| = |\tilde{v}_l|$.*

*Proof.* For every $l \in [k]$, let $\Delta_l = \|\mathbf{w}_l(0)\|^2 - v_l(0)^2$. By Lemma C.3, we have for every $l \in [k]$ and $t \geq 0$ that $\|\mathbf{w}_l(t)\|^2 - v_l(t)^2 = \Delta_l$, namely, the differences between the square norms of the incoming and outgoing weights of each hidden neuron remain constant during the training. Hence, we have

$$|\tilde{v}_l| = \|\tilde{\boldsymbol{\theta}}\| \cdot \lim_{t\to\infty}\frac{|v_l(t)|}{\|\boldsymbol{\theta}(t)\|} = \|\tilde{\boldsymbol{\theta}}\| \cdot \lim_{t\to\infty}\frac{\sqrt{\|\mathbf{w}_l(t)\|^2 - \Delta_l}}{\|\boldsymbol{\theta}(t)\|}.$$

Thus, if $\lim_{t\to\infty}\|\mathbf{w}_l(t)\| = \infty$, then we have $|\tilde{v}_l| = \|\tilde{\boldsymbol{\theta}}\| \cdot \lim_{t\to\infty}\frac{\|\mathbf{w}_l(t)\|}{\|\boldsymbol{\theta}(t)\|} = \|\tilde{\mathbf{w}}_l\|$.

Assume now that $\|\mathbf{w}_l(t)\| \not\to \infty$. By the definition of $\tilde{\boldsymbol{\theta}}$ we have $\|\tilde{\mathbf{w}}_l\| = \|\tilde{\boldsymbol{\theta}}\| \cdot \lim_{t\to\infty}\frac{\|\mathbf{w}_l(t)\|}{\|\boldsymbol{\theta}(t)\|}$. Since $\lim_{t\to\infty}\frac{\|\mathbf{w}_l(t)\|}{\|\boldsymbol{\theta}(t)\|}$ exists and $\lim_{t\to\infty}\|\boldsymbol{\theta}(t)\| = \infty$, then we have $\lim_{t\to\infty}\frac{\|\mathbf{w}_l(t)\|}{\|\boldsymbol{\theta}(t)\|} = 0$. Hence, $\lim_{t\to\infty}\frac{|v_l(t)|}{\|\boldsymbol{\theta}(t)\|} = \lim_{t\to\infty}\frac{\sqrt{\|\mathbf{w}_l(t)\|^2 - \Delta_l}}{\|\boldsymbol{\theta}(t)\|} = 0$. Therefore $\|\tilde{\mathbf{w}}_l\| = \tilde{v}_l = 0$. $\qquad\square$

### C.2 Proof of Theorem 3.1

Suppose that the network $\Phi$ is parameterized by $\boldsymbol{\theta} = [W^{(1)}, \ldots, W^{(m)}]$. By Theorem 2.1, GF converges in direction to a KKT point $\tilde{\boldsymbol{\theta}} = [\tilde{W}^{(1)}, \ldots, \tilde{W}^{(m)}]$ of Problem 2. For every $l \in [m]$ let

$\Delta_l = \left\|W^{(l)}(0)\right\|_F^2 - \left\|W^{(1)}(0)\right\|_F^2$. By Lemma C.1, we have for every $l \in [m]$ and $t \geq 0$ that

$$\left\|W^{(l)}(t)\right\|_F^2 - \left\|W^{(1)}(t)\right\|_F^2 = \sum_{j=1}^{l-1} \left\|W^{(j+1)}(t)\right\|_F^2 - \left\|W^{(j)}(t)\right\|_F^2$$

$$= \sum_{j=1}^{l-1} \left\|W^{(j+1)}(0)\right\|_F^2 - \left\|W^{(j)}(0)\right\|_F^2$$

$$= \left\|W^{(l)}(0)\right\|_F^2 - \left\|W^{(1)}(0)\right\|_F^2 = \Delta_l .$$

Hence, we have

$$\left\|\tilde{W}^{(l)}\right\|_F = \|\tilde{\boldsymbol{\theta}}\| \cdot \lim_{t\to\infty} \frac{\left\|W^{(l)}(t)\right\|_F}{\|\boldsymbol{\theta}(t)\|} = \|\tilde{\boldsymbol{\theta}}\| \cdot \lim_{t\to\infty} \frac{\sqrt{\left\|W^{(1)}(t)\right\|_F^2 + \Delta_l}}{\|\boldsymbol{\theta}(t)\|} .$$

Since by Theorem 2.1 we have $\lim_{t\to\infty} \|\boldsymbol{\theta}(t)\| = \infty$, then $\lim_{t\to\infty} \left\|W^{(1)}(t)\right\|_F = \infty$, and we have

$$\left\|\tilde{W}^{(l)}\right\|_F = \|\tilde{\boldsymbol{\theta}}\| \cdot \lim_{t\to\infty} \frac{\left\|W^{(1)}(t)\right\|_F}{\|\boldsymbol{\theta}(t)\|} = \left\|\tilde{W}^{(1)}\right\|_F := C .$$

By Ji and Telgarsky [2020] (Proposition 4.4), when GF on a fully-connected linear network w.r.t. the exponential loss or the logistic loss converges to zero loss, then we have the following. There are unit vectors $\mathbf{v}_0, \ldots, \mathbf{v}_m$ such that

$$\lim_{t\to\infty} \frac{W^{(l)}(t)}{\left\|W^{(l)}(t)\right\|_F} = \mathbf{v}_l \mathbf{v}_{l-1}^\top$$

for every $l \in [m]$. Moreover, we have $\mathbf{v}_m = 1$, and $\mathbf{v}_0 = \mathbf{u}$ where

$$\mathbf{u} := \operatorname*{argmax}_{\|\mathbf{u}\|=1} \min_{i\in[n]} y_i \mathbf{u}^\top \mathbf{x}_i$$

is the unique linear max margin predictor.

Note that we have

$$\frac{\tilde{W}^{(l)}}{C} = \frac{\tilde{W}^{(l)}}{\left\|\tilde{W}^{(l)}\right\|_F} = \frac{\|\tilde{\boldsymbol{\theta}}\| \cdot \lim_{t\to\infty} \frac{W^{(l)}(t)}{\|\boldsymbol{\theta}(t)\|}}{\|\tilde{\boldsymbol{\theta}}\| \cdot \lim_{t\to\infty} \frac{\left\|W^{(l)}(t)\right\|_F}{\|\boldsymbol{\theta}(t)\|}} = \lim_{t\to\infty} \frac{W^{(l)}(t)}{\left\|W^{(l)}(t)\right\|_F} = \mathbf{v}_l \mathbf{v}_{l-1}^\top .$$

Thus, $\tilde{W}^{(l)} = C \mathbf{v}_l \mathbf{v}_{l-1}^\top$ for every $l \in [m]$.

Let $\tilde{\mathbf{u}} = \tilde{W}^{(m)} \cdot \ldots \cdot \tilde{W}^{(1)} = C^m \mathbf{u}$. Since $\tilde{\boldsymbol{\theta}}$ is a KKT point of Problem 2, we have for every $l \in [m]$

$$\tilde{W}^{(l)} = \sum_{i\in[n]} \lambda_i y_i \frac{\partial \Phi(\tilde{\boldsymbol{\theta}}; \mathbf{x}_i)}{\partial W^{(l)}} ,$$

where $\lambda_i \geq 0$ for every $i$, and $\lambda_i = 0$ if $y_i \Phi(\tilde{\boldsymbol{\theta}}; \mathbf{x}_i) \neq 1$. Since $\tilde{W}^{(l)}$ are non-zero then there is $i \in [n]$ such that $1 = y_i \Phi(\tilde{\boldsymbol{\theta}}; \mathbf{x}_i) = y_i \tilde{\mathbf{u}}^\top \mathbf{x}_i = y_i C^m \mathbf{u}^\top \mathbf{x}_i$. Likewise, since $\tilde{\boldsymbol{\theta}}$ satisfies the constraints of Problem 2, then for every $i \in [n]$ we have $1 \leq y_i \Phi(\tilde{\boldsymbol{\theta}}; \mathbf{x}_i) = y_i C^m \mathbf{u}^\top \mathbf{x}_i$. Since $\mathbf{u}$ is a unit vector that maximized the margin, then we have

$$\|\tilde{\mathbf{u}}\| = C^m = \min \|\mathbf{u}'\| \text{ s.t. } y_i \mathbf{u}'^\top \mathbf{x}_i \geq 1 \text{ for all } i \in [n] . \tag{6}$$

Assume toward contradiction that there is $\boldsymbol{\theta}'$ with $\|\boldsymbol{\theta}'\| < \|\tilde{\boldsymbol{\theta}}\|$ that satisfies the constraints in Problem 2. Let $\mathbf{u}' = W'^{(m)} \cdot \ldots \cdot W'^{(1)}$. By Eq. 6 we have $\|\mathbf{u}'\| \geq \|\tilde{\mathbf{u}}\| = C^m$. Moreover, we have $\|\mathbf{u}'\| = \left\|W'^{(m)} \cdot \ldots \cdot W'^{(1)}\right\| \leq \prod_{l\in[m]} \left\|W'^{(l)}\right\|_F$ due to the submultiplicativity of the Frobenius norm. Hence $\prod_{l\in[m]} \left\|W'^{(l)}\right\|_F \geq C^m$. The following lemma implies that

$$\|\boldsymbol{\theta}'\|^2 = \sum_{l\in[m]} \left\|W'^{(l)}\right\|_F^2 \geq m \cdot C^2 = \sum_{l\in[m]} \left\|\tilde{W}^{(l)}\right\|_F^2 = \left\|\tilde{\boldsymbol{\theta}}\right\|^2$$

in contradiction to our assumption, and thus completes the proof.

**Lemma C.5.** *Let $a_1, \ldots, a_m$ be real numbers such that $\prod_{j \in [m]} a_j \geq C^m$ for some $C \geq 0$. Then $\sum_{j \in [m]} a_j^2 \geq m \cdot C^2$.*

*Proof.* It suffices to prove the claim for the case where $\prod_{j \in [m]} a_j = C^m$. Indeed, if $\prod_{j \in [m]} a_j > C^m$ then we can replace some $a_j$ with an appropriate $a_j'$ such that $|a_j'| < |a_j|$ and we only decrease $\sum_{j \in [m]} a_j^2$. Consider the following problem

$$\min \frac{1}{2} \sum_{j \in [m]} a_j^2 \quad \text{s.t.} \quad \prod_{j \in [m]} a_j = C^m .$$

Using the Lagrange multipliers we obtain that there is some $\lambda \in \mathbb{R}$ such that for every $l \in [m]$ we have $a_l = \lambda \cdot \prod_{j \neq l} a_j$. Thus, $a_l^2 = \lambda \cdot \prod_{j \in [m]} a_j$. It implies that $a_1^2 = \ldots = a_m^2$. Since $\prod_{j \in [m]} a_j = C^m$ then $|a_j| = C$ for every $j \in [m]$. Hence, $\sum_{j \in [m]} a_j^2 = mC^2$. $\qquad\square$

### C.3 Proof of Theorem 3.2

Consider an initialization $\boldsymbol{\theta}(0)$ is such that $\mathbf{w}_1(0)$ satisfies $\langle \mathbf{w}_1(0), \mathbf{x}_1 \rangle > 0$ and $\langle \mathbf{w}_1(0), \mathbf{x}_2 \rangle > 0$, and $\mathbf{w}_2(0)$ satisfies $\langle \mathbf{w}_2(0), \mathbf{x}_1 \rangle < 0$ and $\langle \mathbf{w}_2(0), \mathbf{x}_2 \rangle < 0$. Moreover, assume that $v_1(0) > 0$.

Note that for every $\boldsymbol{\theta}$ such that $\langle \mathbf{w}_2, \mathbf{x}_1 \rangle < 0$ and $\langle \mathbf{w}_2, \mathbf{x}_2 \rangle < 0$ we have

$$\begin{aligned}
\nabla_{\mathbf{w}_2} \mathcal{L}(\boldsymbol{\theta}) &= \sum_{i=1}^{2} \ell'(y_i \Phi(\boldsymbol{\theta}; \mathbf{x}_i)) \cdot y_i \nabla_{\mathbf{w}_2} \Phi(\boldsymbol{\theta}; \mathbf{x}_i) \\
&= \sum_{i=1}^{2} \ell'(y_i \Phi(\boldsymbol{\theta}; \mathbf{x}_i)) \cdot y_i \nabla_{\mathbf{w}_2} \left[ v_1 \sigma(\mathbf{w}_1^\top \mathbf{x}_i) + v_2 \sigma(\mathbf{w}_2^\top \mathbf{x}_i) \right] \\
&= \sum_{i=1}^{2} \ell'(y_i \Phi(\boldsymbol{\theta}; \mathbf{x}_i)) \cdot y_i v_2 \sigma'(\mathbf{w}_2^\top \mathbf{x}_i) \mathbf{x}_i = \mathbf{0} .
\end{aligned}$$

and

$$\begin{aligned}
\nabla_{v_2} \mathcal{L}(\boldsymbol{\theta}) &= \sum_{i=1}^{2} \ell'(y_i \Phi(\boldsymbol{\theta}; \mathbf{x}_i)) \cdot y_i \nabla_{v_2} \Phi(\boldsymbol{\theta}; \mathbf{x}_i) \\
&= \sum_{i=1}^{2} \ell'(y_i \Phi(\boldsymbol{\theta}; \mathbf{x}_i)) \cdot y_i \nabla_{v_2} \left[ v_1 \sigma(\mathbf{w}_1^\top \mathbf{x}_i) + v_2 \sigma(\mathbf{w}_2^\top \mathbf{x}_i) \right] \\
&= \sum_{i=1}^{2} \ell'(y_i \Phi(\boldsymbol{\theta}; \mathbf{x}_i)) \cdot y_i \sigma(\mathbf{w}_2^\top \mathbf{x}_i) = 0 .
\end{aligned}$$

Hence, $\mathbf{w}_2$ and $v_2$ get stuck in their initial values. Moreover, we have

$$\begin{aligned}
\nabla_{v_1} \mathcal{L}(\boldsymbol{\theta}) &= \sum_{i=1}^{2} \ell'(y_i \Phi(\boldsymbol{\theta}; \mathbf{x}_i)) \cdot y_i \nabla_{v_1} \left[ v_1 \sigma(\mathbf{w}_1^\top \mathbf{x}_i) + v_2 \sigma(\mathbf{w}_2^\top \mathbf{x}_i) \right] \\
&= \sum_{i=1}^{2} \ell'(y_i \Phi(\boldsymbol{\theta}; \mathbf{x}_i)) \cdot \sigma(\mathbf{w}_1^\top \mathbf{x}_i) \leq 0 .
\end{aligned}$$

Therefore, for every $t \geq 0$ we have $v_1(t) \geq v_1(0) > 0$.

We denote $\mathbf{w}_1 = (w_1[1], w_1[2])$. Since $\langle \mathbf{w}_1(0), \mathbf{x}_j \rangle > 0$ for $j \in \{1, 2\}$ then $w_1[2](0) > 0$. Assume w.l.o.g. that $w_1[1](0) \geq 0$ (the case where $w_1[1](0) \leq 0$ is similar). For every $\mathbf{w}_1$ that satisfies

$w_1[2] \geq 0$ and $0 \leq w_1[1] \leq w_1[1](0)$ we have $\langle \mathbf{w}_1, \mathbf{x}_1 \rangle > \langle \mathbf{w}_1, \mathbf{x}_2 \rangle > 0$. Thus,

$$
\begin{aligned}
\nabla_{\mathbf{w}_1} \mathcal{L}(\boldsymbol{\theta}) &= \sum_{i=1}^{2} \ell'(y_i \Phi(\boldsymbol{\theta}; \mathbf{x}_i)) \cdot y_i \nabla_{\mathbf{w}_1} \left[ v_1 \sigma(\mathbf{w}_1^\top \mathbf{x}_i) + v_2 \sigma(\mathbf{w}_2^\top \mathbf{x}_i) \right] \\
&= \sum_{i=1}^{2} \ell'(y_i (v_1 \sigma(\mathbf{w}_1^\top \mathbf{x}_i) + 0)) \cdot y_i v_1 \sigma'(\mathbf{w}_1^\top \mathbf{x}_i) \mathbf{x}_i \\
&= \sum_{i=1}^{2} \ell'(v_1 \mathbf{w}_1^\top \mathbf{x}_i) \cdot v_1 \mathbf{x}_i .
\end{aligned}
$$

Since $\ell'$ is negative and monotonically increasing, and since $v_1 \mathbf{w}_1^\top \mathbf{x}_1 > v_1 \mathbf{w}_1^\top \mathbf{x}_2$, then $\frac{dw_1[1]}{dt} \leq 0$. Also, $\frac{dw_1[2]}{dt} > 0$. Moreover, if $w_1[1] = 0$ then $v_1 \mathbf{w}_1^\top \mathbf{x}_1 = v_1 \mathbf{w}_1^\top \mathbf{x}_2$ and thus $\frac{dw_1[1]}{dt} = 0$. Hence, for every $t$ we have $w_1[2](t) \geq w_1[2](0) > 0$ and $0 \leq w_1[1](t) \leq w_1[1](0)$.

If $\mathcal{L}(\boldsymbol{\theta}) \geq 1$ then for some $i \in \{1, 2\}$ we have $\ell(y_i \Phi(\boldsymbol{\theta}; \mathbf{x}_i)) \geq \frac{1}{2}$ and hence $\ell'(y_i \Phi(\boldsymbol{\theta}; \mathbf{x}_i)) \leq c$ for some constant $c < 0$. Since we also have $v_1 \geq v_1(0) > 0$, we have

$$
\frac{dw_1[2]}{dt} \geq -c \cdot v_1(0) \cdot \frac{1}{4} .
$$

Therefore, if the initialization $\boldsymbol{\theta}(0)$ is such that $\mathcal{L}(\boldsymbol{\theta}) \geq 1$ then $w_1[2](t)$ increases at rate at least $\frac{(-c) \cdot v_1(0)}{4}$ while $w_1[1](t)$ remains in $[0, w_1[1](0)]$. Note that for such $w_1[1]$ and $v_1 \geq v_1(0) > 0$, if $w_1[2]$ is sufficiently large then we have $v_1 \langle \mathbf{w}_1, \mathbf{x}_i \rangle \geq 1$ for $i \in \{1, 2\}$. Hence, there is some $t_0$ such that $\mathcal{L}(\boldsymbol{\theta}(t_0)) \leq 2\ell(1) < 1$ for both the exponential loss and the logistic loss.

Therefore, by Theorem 2.1 GF converges in direction to a KKT point of Problem 2, and we have $\lim_{t \to \infty} \mathcal{L}(\boldsymbol{\theta}(t)) = 0$ and $\lim_{t \to \infty} \|\boldsymbol{\theta}(t)\| = \infty$. It remains to show that it does not converge in direction to a local optimum of Problem 2.

Let $\bar{\boldsymbol{\theta}} = \lim_{t \to \infty} \frac{\boldsymbol{\theta}(t)}{\|\boldsymbol{\theta}(t)\|}$. We denote $\bar{\boldsymbol{\theta}} = [\bar{\mathbf{w}}_1, \bar{\mathbf{w}}_2, \bar{v}_1, \bar{v}_2]$. We show that $\bar{\mathbf{w}}_1 = \frac{1}{\sqrt{2}}(0, 1)^\top$, $\bar{v}_1 = \frac{1}{\sqrt{2}}$, $\bar{\mathbf{w}}_2 = \mathbf{0}$ and $\bar{v}_2 = 0$. By Lemma C.1, we have for every $t \geq 0$ that $v_1(t)^2 - \|\mathbf{w}_1(t)\|^2 = v_1(0)^2 - \|\mathbf{w}_1(0)\|^2 := \Delta$. Since for every $t$ we have $\mathbf{w}_2(t) = \mathbf{w}_2(0)$ and $v_2(t) = v_2(0)$, and since $\lim_{t \to \infty} \|\boldsymbol{\theta}(t)\| = \infty$ then we have $\lim_{t \to \infty} \|\mathbf{w}_1(t)\| = \infty$ and $\lim_{t \to \infty} |v_1(t)| = \infty$. Also, since $\lim_{t \to \infty} \|\mathbf{w}_1(t)\| = \infty$ and $w_1[1](t) \in [0, w_1[1](0)]$ then $\lim_{t \to \infty} w_1[2](t) = \infty$. Note that

$$
\|\boldsymbol{\theta}(t)\| = \sqrt{\|\mathbf{w}_1(t)\|^2 + v_1(t)^2 + \|\mathbf{w}_2(0)\|^2 + v_2(0)^2} = \sqrt{\Delta + 2\|\mathbf{w}_1(t)\|^2 + \|\mathbf{w}_2(0)\|^2 + v_2(0)^2} .
$$

Since $w_1[1](t) \in [0, w_1[1](0)]$ and $\|\boldsymbol{\theta}(t)\| \to \infty$, we have

$$
\bar{\mathbf{w}}_1[1] = \lim_{t \to \infty} \frac{w_1[1](t)}{\|\boldsymbol{\theta}(t)\|} = 0 .
$$

Moreover,

$$
\bar{\mathbf{w}}_1[2] = \lim_{t \to \infty} \frac{w_1[2](t)}{\|\boldsymbol{\theta}(t)\|} = \lim_{t \to \infty} \sqrt{\frac{(w_1[2](t))^2}{\Delta + 2(w_1[1](t))^2 + 2(w_1[2](t))^2 + \|\mathbf{w}_2(0)\|^2 + v_2(0)^2}} = \frac{1}{\sqrt{2}} ,
$$

and

$$
\bar{\mathbf{w}}_2 = \lim_{t \to \infty} \frac{\mathbf{w}_2(t)}{\|\boldsymbol{\theta}(t)\|} = \lim_{t \to \infty} \frac{\mathbf{w}_2(0)}{\|\boldsymbol{\theta}(t)\|} = \mathbf{0} .
$$

Finally, by Lemma C.4 and since $v_1(t) > 0$, we have $\bar{v}_1 = \|\bar{\mathbf{w}}_1\| = \frac{1}{\sqrt{2}}$. By Lemma C.4 we also have $|\bar{v}_2| = \|\bar{\mathbf{w}}_2\| = 0$.

Next, we show that $\bar{\boldsymbol{\theta}}$ does not point at the direction of a local optimum of Problem 2. Let $\tilde{\boldsymbol{\theta}} = [\tilde{\mathbf{w}}_1, \tilde{\mathbf{w}}_2, \tilde{v}_1, \tilde{v}_2]$ be a KKT point of Problem 2 that points at the direction of $\bar{\boldsymbol{\theta}}$. Such $\tilde{\boldsymbol{\theta}}$ exists since $\boldsymbol{\theta}(t)$ converges in direction to a KKT point. Thus, we have $\tilde{\mathbf{w}}_2 = \mathbf{0}$, $\tilde{v}_2 = 0$, $\tilde{\mathbf{w}}_1 = \alpha(0, 1)^\top$ and $\tilde{v}_1 = \alpha$ for some $\alpha > 0$. Since $\tilde{\boldsymbol{\theta}}$ satisfies the KKT conditions, we have

$$
\tilde{\mathbf{w}}_1 = \sum_{i=1}^{2} \lambda_i \nabla_{\mathbf{w}_1} \left( y_i \Phi(\tilde{\boldsymbol{\theta}}; \mathbf{x}_i) \right) = \sum_{i=1}^{2} \lambda_i y_i \left( \tilde{v}_1 \sigma'(\tilde{\mathbf{w}}_1^\top \mathbf{x}_i) \mathbf{x}_i \right) ,
$$

where $\lambda_i \geq 0$ and $\lambda_i = 0$ if $y_i \Phi(\tilde{\boldsymbol{\theta}}; \mathbf{x}_i) \neq 1$. Note that the KKT condition should be w.r.t. the Clarke subdifferential, but since $\tilde{\mathbf{w}}_1^\top \mathbf{x}_i > 0$ for $i \in \{1, 2\}$ then we use here the gradient. Hence, there is $i \in \{1, 2\}$ such that $y_i \Phi(\tilde{\boldsymbol{\theta}}; \mathbf{x}_i) = 1$. Thus,

$$1 = y_i \Phi(\tilde{\boldsymbol{\theta}}; \mathbf{x}_i) = \tilde{v}_1 \sigma(\tilde{\mathbf{w}}_1^\top \mathbf{x}_i) + \tilde{v}_2 \sigma(\tilde{\mathbf{w}}_2^\top \mathbf{x}_i) = \alpha \cdot \frac{\alpha}{4} + 0 = \frac{\alpha^2}{4} .$$

Therefore, $\alpha = 2$ and we have $\tilde{\mathbf{w}}_1 = (0, 2)^\top$ and $\tilde{v}_1 = 2$.

In order to show that $\tilde{\boldsymbol{\theta}}$ is not a local optimum, we show that for every $0 < \epsilon' < 1$ there exists some $\boldsymbol{\theta}'$ such that $\left\| \boldsymbol{\theta}' - \tilde{\boldsymbol{\theta}} \right\| \leq \epsilon'$, $\boldsymbol{\theta}'$ satisfies $\Phi(\boldsymbol{\theta}'; \mathbf{x}_i) \geq 1$ for every $i \in \{1, 2\}$, and $\|\boldsymbol{\theta}'\| < \|\tilde{\boldsymbol{\theta}}\|$. Let $\epsilon = \frac{\epsilon'^2}{9} < \frac{1}{2}$. Let $\boldsymbol{\theta}' = [\mathbf{w}_1', \mathbf{w}_2', v_1', v_2']$ be such that $\mathbf{w}_1' = (\frac{\epsilon}{2}, 2 - 2\epsilon)^\top$, $\mathbf{w}_2' = (-\sqrt{2\epsilon}, 0)^\top$, $v_1' = 2$ and $v_2' = \sqrt{2\epsilon}$. Note that

$$\Phi(\boldsymbol{\theta}'; \mathbf{x}_1) = 2 \cdot \sigma\left( (\frac{\epsilon}{2}, 2 - 2\epsilon)(1, \frac{1}{4})^\top \right) + \sqrt{2\epsilon} \cdot \sigma\left( (-\sqrt{2\epsilon}, 0)(1, \frac{1}{4})^\top \right)$$

$$= 2 \cdot \sigma\left( \frac{\epsilon}{2} + \frac{1}{2} - \frac{\epsilon}{2} \right) + \sqrt{2\epsilon} \cdot \sigma\left( -\sqrt{2\epsilon} \right) = 1 ,$$

and

$$\Phi(\boldsymbol{\theta}'; \mathbf{x}_2) = 2 \cdot \sigma\left( (\frac{\epsilon}{2}, 2 - 2\epsilon)(-1, \frac{1}{4})^\top \right) + \sqrt{2\epsilon} \cdot \sigma\left( (-\sqrt{2\epsilon}, 0)(-1, \frac{1}{4})^\top \right)$$

$$= 2 \cdot \sigma\left( -\frac{\epsilon}{2} + \frac{1}{2} - \frac{\epsilon}{2} \right) + \sqrt{2\epsilon} \cdot \sigma\left( \sqrt{2\epsilon} \right) = 1 - 2\epsilon + 2\epsilon = 1 .$$

We also have

$$\left\| \boldsymbol{\theta}' - \tilde{\boldsymbol{\theta}} \right\|^2 = \|\mathbf{w}_1' - \tilde{\mathbf{w}}_1\|^2 + \|\mathbf{w}_2' - \tilde{\mathbf{w}}_2\|^2 + (v_1' - \tilde{v}_1)^2 + (v_2' - \tilde{v}_2)^2$$

$$= \left( \frac{\epsilon^2}{4} + 4\epsilon^2 \right) + 2\epsilon + 0 + 2\epsilon < 9\epsilon = \epsilon'^2 .$$

Finally, we have

$$\|\boldsymbol{\theta}'\|^2 = \frac{\epsilon^2}{4} + 4 - 8\epsilon + 4\epsilon^2 + 2\epsilon + 4 + 2\epsilon = 8 - 4\epsilon + \frac{17\epsilon^2}{4} < 8 - 4\epsilon + \frac{17\epsilon}{8} < 8 = \left\| \tilde{\boldsymbol{\theta}} \right\|^2 .$$

Thus, $\|\boldsymbol{\theta}'\| < \|\tilde{\boldsymbol{\theta}}\|$.

### C.4 Proof of Theorem 4.1

Let $\mathbf{x} = (1, 2)^\top$ and $y = 1$. Let $\boldsymbol{\theta}(0)$ such that $\mathbf{w}_1(0) = \mathbf{w}_2(0) = (1, 0)^\top$. Note that $\mathcal{L}(\boldsymbol{\theta}(0)) = \ell(1) < 1$ for both linear and ReLU networks with the exponential loss or the logistic loss, and therefore by Theorem 2.1 GF converges in direction to a KKT point $\tilde{\boldsymbol{\theta}}$ of Problem 2, and we have $\lim_{t \to \infty} \mathcal{L}(\boldsymbol{\theta}(t)) = 0$ and $\lim_{t \to \infty} \|\boldsymbol{\theta}(t)\| = \infty$. We denote $\mathbf{w}_1 = (\mathbf{w}_1[1], \mathbf{w}_1[2])^\top$ and $\mathbf{w}_2 = (\mathbf{w}_2[1], \mathbf{w}_2[2])^\top$. Note that the initialization $\boldsymbol{\theta}(0)$ is such that the second hidden neuron has $0$ in both its incoming and outgoing weights. Hence, the gradient w.r.t. $\mathbf{w}_1[2]$ and $\mathbf{w}_2[2]$ is zero, and the second hidden neuron remains inactive during the training. Moreover, $\mathbf{w}_1[1]$ and $\mathbf{w}_2[1]$ are strictly increasing. Also, by Lemma C.3 we have for every $t \geq 0$ that $\mathbf{w}_1[1](t)^2 = \mathbf{w}_2[1](t)^2$. Overall, $\tilde{\boldsymbol{\theta}}$ is such that $\tilde{\mathbf{w}}_1 = \tilde{\mathbf{w}}_2 = (1, 0)^\top$. Note that since the dataset is of size 1, then every KKT point of Problem 2 must label the input $\mathbf{x}$ with exactly 1.

It remains to show that $\tilde{\boldsymbol{\theta}}$ is not local optimum. Let $0 < \epsilon < 1$, and let $\boldsymbol{\theta}' = [\mathbf{w}_1', \mathbf{w}_2']$ with $\mathbf{w}_1' = \mathbf{w}_2' = \left( \sqrt{1 - \epsilon}, \sqrt{\frac{\epsilon}{2}} \right)^\top$. Note that $\boldsymbol{\theta}'$ satisfies the constraints of Problem 2, since $y \cdot \Phi(\boldsymbol{\theta}'; \mathbf{x}) = 1 - \epsilon + 2 \cdot \frac{\epsilon}{2} = 1$. Moreover, we have $\|\tilde{\boldsymbol{\theta}}\|^2 = 2$ and $\|\boldsymbol{\theta}'\|^2 = 2\left( 1 - \epsilon + \frac{\epsilon}{2} \right) = 2 - \epsilon$ and therefore $\|\boldsymbol{\theta}'\| < \|\tilde{\boldsymbol{\theta}}\|$.

## C.5 Proof of Theorem 4.2

### C.5.1 Proof of part 1

We assume w.l.o.g. that the second layer is fully-connected, namely, all hidden neurons are connected to the output neuron, since otherwise we can ignore disconnected neurons. For the network $\Phi$ we use the parameterization $\boldsymbol{\theta} = [\mathbf{w}_1, \ldots, \mathbf{w}_k, \mathbf{v}]$ introduced in Section C.1. Thus, we have $\Phi(\boldsymbol{\theta}; \mathbf{x}) = \sum_{l \in [k]} v_l \mathbf{w}_l^\top \mathbf{x}^l$.

By Theorem 2.1, GF converges in direction to $\tilde{\boldsymbol{\theta}} = [\tilde{\mathbf{w}}_1, \ldots, \tilde{\mathbf{w}}_k, \tilde{\mathbf{v}}]$ which satisfies the KKT conditions of Problem 2. Thus, there are $\lambda_1, \ldots, \lambda_n$ such that for every $j \in [k]$ we have

$$\tilde{\mathbf{w}}_j = \sum_{i \in [n]} \lambda_i \nabla_{\mathbf{w}_j} \left( y_i \Phi(\tilde{\boldsymbol{\theta}}; \mathbf{x}_i) \right) = \sum_{i \in [n]} \lambda_i y_i \tilde{v}_j \mathbf{x}_i^j \,, \tag{7}$$

and we have $\lambda_i \geq 0$ for all $i$, and $\lambda_i = 0$ if $y_i \Phi(\tilde{\boldsymbol{\theta}}; \mathbf{x}_i) = y_i \sum_{l \in [k]} \tilde{v}_l \tilde{\mathbf{w}}_l^\top \mathbf{x}_i^l \neq 1$. By Theorem 2.1, we also have $\lim_{t \to \infty} \|\boldsymbol{\theta}(t)\| = \infty$. Hence, by Lemma C.4 we have $\|\tilde{\mathbf{w}}_j\| = |\tilde{v}_j|$ for all $j \in [k]$.

Consider the following problem

$$\min \sum_{l \in [k]} \|\mathbf{u}_l\| \quad \text{s.t.} \quad \forall i \in [n] \; y_i \sum_{l \in [k]} \mathbf{u}_l^\top \mathbf{x}_i^l \geq 1 \,. \tag{8}$$

For every $l \in [k]$ we denote $\tilde{\mathbf{u}}_l = \tilde{v}_l \cdot \tilde{\mathbf{w}}_l$. Since we assume that $\tilde{\mathbf{w}}_l \neq \mathbf{0}$ for every $l \in [k]$, and since $\|\tilde{\mathbf{w}}_l\| = |\tilde{v}_l|$, then $\tilde{\mathbf{u}}_l \neq \mathbf{0}$ for all $l \in [k]$. Note that since $\tilde{\mathbf{w}}_1, \ldots, \tilde{\mathbf{w}}_k, \tilde{\mathbf{v}}$ satisfy the constraints in Problem 2, then $\tilde{\mathbf{u}}_1, \ldots, \tilde{\mathbf{u}}_k$ satisfy the constraints in the above problem. In order to show that $\tilde{\mathbf{u}}_1, \ldots, \tilde{\mathbf{u}}_k$ satisfy the KKT condition of the problem, we need to prove that for every $j \in [k]$ we have

$$\frac{\tilde{\mathbf{u}}_j}{\|\tilde{\mathbf{u}}_j\|} = \sum_{i \in [n]} \lambda_i' y_i \mathbf{x}_i^j \tag{9}$$

for some $\lambda_i' \geq 0$ such that $\lambda_i' = 0$ if $y_i \sum_{l \in [k]} \tilde{\mathbf{u}}_l^\top \mathbf{x}_i^l \neq 1$. From Eq. 7 and since $\|\tilde{\mathbf{w}}_l\| = |\tilde{v}_l|$ for every $l \in [k]$, we have

$$\tilde{\mathbf{u}}_j = \tilde{v}_j \cdot \tilde{\mathbf{w}}_j = \tilde{v}_j \sum_{i \in [n]} \lambda_i y_i \tilde{v}_j \mathbf{x}_i^j = \tilde{v}_j^2 \sum_{i \in [n]} \lambda_i y_i \mathbf{x}_i^j = \|\tilde{v}_j \tilde{\mathbf{w}}_j\| \sum_{i \in [n]} \lambda_i y_i \mathbf{x}_i^j = \|\tilde{\mathbf{u}}_j\| \sum_{i \in [n]} \lambda_i y_i \mathbf{x}_i^j \,.$$

Note that we have $\lambda_i \geq 0$ for all $i$, and $\lambda_i = 0$ if $y_i \sum_{l \in [k]} \tilde{\mathbf{u}}_l^\top \mathbf{x}_i^l = y_i \sum_{l \in [k]} \tilde{v}_l \tilde{\mathbf{w}}_l^\top \mathbf{x}_i^l \neq 1$. Hence Eq. 9 holds with $\lambda_1', \ldots, \lambda_n'$ that satisfy the requirement. Since the objective in Problem 8 is convex and the constraints are affine functions, then its KKT condition is sufficient for global optimality. Namely, $\tilde{\mathbf{u}}_1, \ldots, \tilde{\mathbf{u}}_k$ are a global optimum for problem 8.

We now deduce that $\tilde{\boldsymbol{\theta}}$ is a global optimum for Problem 2. Assume toward contradiction that there is a solution $\boldsymbol{\theta}' = [\mathbf{w}_1', \ldots, \mathbf{w}_k', \mathbf{v}']$ for the constraints in Problem 2 such that $\|\boldsymbol{\theta}'\|^2 < \|\tilde{\boldsymbol{\theta}}\|^2$. Let $\mathbf{u}_l' = v_l' \mathbf{w}_l'$. Note that the vectors $\mathbf{u}_l'$ satisfy the constraints in Problem 8. Moreover, we have

$$\sum_{l \in [k]} \|\mathbf{u}_l'\| = \sum_{l \in [k]} |v_l'| \cdot \|\mathbf{w}_l'\| \leq \sum_{l \in [k]} \frac{1}{2} \left( |v_l'|^2 + \|\mathbf{w}_l'\|^2 \right) = \frac{1}{2} \|\boldsymbol{\theta}'\|^2 < \frac{1}{2} \left\| \tilde{\boldsymbol{\theta}} \right\|^2$$

$$= \sum_{l \in [k]} \frac{1}{2} \left( |\tilde{v}_l|^2 + \|\tilde{\mathbf{w}}_l\|^2 \right) \,.$$

Since $\|\tilde{\mathbf{w}}_l\| = |\tilde{v}_l|$, the above equals

$$\sum_{l \in [k]} \|\tilde{\mathbf{w}}_l\|^2 = \sum_{l \in [k]} |\tilde{v}_l| \cdot \|\tilde{\mathbf{w}}_l\| = \sum_{l \in [k]} \|\tilde{\mathbf{u}}_l\| \,,$$

which contradicts the global optimality of $\tilde{\mathbf{u}}_1, \ldots, \tilde{\mathbf{u}}_k$.

### C.5.2 Proof of part 2

Let $\{(\mathbf{x}_i, y_i)\}_{i=1}^4$ be a dataset such that $y_i = 1$ for all $i \in [4]$ and we have $\mathbf{x}_1 = (0, 1)^\top$, $\mathbf{x}_2 = (1, 0)^\top$, $\mathbf{x}_3 = (0, -1)$ and $\mathbf{x}_4 = (-1, 0)$. Consider the initialization $\boldsymbol{\theta}(0) = [\mathbf{w}_1(0), \mathbf{w}_2(0), \mathbf{w}_3(0), \mathbf{w}_4(0), \mathbf{v}(0)]$ such that $\mathbf{w}_i(0) = 2\mathbf{x}_i$ and $v_i(0) = 2$ for every $i \in [4]$. Note that $\mathcal{L}(\boldsymbol{\theta}(0)) = 4\ell(4) < 1$ for both the exponential loss and the logistic loss, and therefore by Theorem 2.1 GF converges in direction to a KKT point $\tilde{\boldsymbol{\theta}}$ of Problem 2, and we have $\lim_{t \to \infty} \mathcal{L}(\boldsymbol{\theta}(t)) = 0$ and $\lim_{t \to \infty} \|\boldsymbol{\theta}(t)\| = \infty$.

We now show that for all $t \geq 0$ we have $\mathbf{w}_i(t) = \alpha(t)\mathbf{x}_i$ and $v_i(t) = \alpha(t)$ where $\alpha(t) > 0$ and $\lim_{t \to \infty} \alpha(t) = \infty$. Indeed, for such $\boldsymbol{\theta}(t)$, for every $j \in [4]$ we have

$$
\begin{aligned}
-\frac{d\mathbf{w}_j}{dt} &= \nabla_{\mathbf{w}_j} \mathcal{L}(\boldsymbol{\theta}) = \sum_{i=1}^4 \ell'(y_i \Phi(\boldsymbol{\theta}; \mathbf{x}_i)) \cdot y_i \nabla_{\mathbf{w}_j} \Phi(\boldsymbol{\theta}; \mathbf{x}_i) \\
&= \sum_{i=1}^4 \ell'\left(\sum_{l=1}^4 v_l \sigma(\mathbf{w}_l^\top \mathbf{x}_i)\right) \cdot \left(v_j \sigma'(\mathbf{w}_j^\top \mathbf{x}_i)\mathbf{x}_i\right) \\
&= \ell'(\alpha^2) \cdot \alpha \cdot \sum_{i=1}^4 \sigma'(\mathbf{w}_j^\top \mathbf{x}_i)\mathbf{x}_i = \ell'(\alpha^2) \cdot \alpha \mathbf{x}_j \;,
\end{aligned}
$$

and

$$
-\frac{dv_j}{dt} = \nabla_{v_j} \mathcal{L}(\boldsymbol{\theta}) = \sum_{i=1}^4 \ell'(y_i \Phi(\boldsymbol{\theta}; \mathbf{x}_i)) \cdot y_i \nabla_{v_j} \Phi(\boldsymbol{\theta}; \mathbf{x}_i) = \sum_{i=1}^4 \ell'\left(\sum_{l=1}^4 v_l \sigma(\mathbf{w}_l^\top \mathbf{x}_i)\right) \cdot \sigma(\mathbf{w}_j^\top \mathbf{x}_i)
$$

$$
= \ell'(\alpha^2) \cdot \sum_{i=1}^4 \sigma(\mathbf{w}_j^\top \mathbf{x}_i) = \ell'(\alpha^2) \cdot \alpha \;.
$$

Moreover, since $\lim_{t \to \infty} \|\boldsymbol{\theta}(t)\| = \infty$ then $\lim_{t \to \infty} \alpha(t) = \infty$.

Hence, the KKT point $\tilde{\boldsymbol{\theta}}$ is such that for every $j \in [4]$ the vector $\tilde{\mathbf{w}}_j$ points at the direction $\mathbf{x}_j$, and we have $\tilde{v}_j = \|\tilde{\mathbf{w}}_j\|$. Also, the vectors $\tilde{\mathbf{w}}_1, \tilde{\mathbf{w}}_2, \tilde{\mathbf{w}}_3, \tilde{\mathbf{w}}_4$ have equal norms. That is, $\tilde{\mathbf{w}}_j = \tilde{\alpha}\mathbf{x}_j$ and $\tilde{v}_j = \tilde{\alpha}$ for some $\tilde{\alpha} > 0$. Moreover, since it satisfies the KKT condition of Problem 2, then we have

$$
\tilde{\mathbf{w}}_j = \sum_{i=1}^4 \lambda_i y_i \nabla_{\mathbf{w}_j} \Phi(\tilde{\boldsymbol{\theta}}; \mathbf{x}_i) \;,
$$

where $\lambda_i \geq 0$ and $\lambda_i = 0$ if $y_i \Phi(\tilde{\boldsymbol{\theta}}; \mathbf{x}_i) \neq 1$. Hence, there is $i$ such that $y_i \Phi(\tilde{\boldsymbol{\theta}}; \mathbf{x}_i) = 1$. Therefore, $\tilde{\alpha}^2 = 1$. Thus, we conclude that for all $j \in [4]$ we have $\tilde{\mathbf{w}}_j = \mathbf{x}_j$ and $\tilde{v}_j = 1$. Note that $\tilde{\mathbf{w}}_j \neq \mathbf{0}$ for all $j \in [4]$ as required.

Next, we show that $\tilde{\boldsymbol{\theta}}$ is not a local optimum of Problem 2. We show that for every $0 < \epsilon < 1$ there exists some $\boldsymbol{\theta}'$ such that $\left\|\boldsymbol{\theta}' - \tilde{\boldsymbol{\theta}}\right\| \leq \epsilon$, $\boldsymbol{\theta}'$ satisfies the constraints of Problem 2, and $\|\boldsymbol{\theta}'\| < \|\tilde{\boldsymbol{\theta}}\|$. Let $\epsilon' = \frac{\epsilon}{2\sqrt{2}}$. Let $\boldsymbol{\theta}'$ be such that $v_j' = \tilde{v}_j = 1$ for all $j \in [4]$, and we have $\mathbf{w}_1' = (\epsilon', 1 - \epsilon')^\top$, $\mathbf{w}_2' = (1 - \epsilon', -\epsilon')^\top$, $\mathbf{w}_3' = (-\epsilon', -1 + \epsilon')^\top$ and $\mathbf{w}_4' = (-1 + \epsilon', \epsilon')^\top$. It is easy to verify that $\boldsymbol{\theta}'$ satisfies the constraints. Indeed, we have $\Phi(\boldsymbol{\theta}'; \mathbf{x}_i) = ((1 - \epsilon') + \epsilon' + 0 + 0) = 1$. Also, we have $\left\|\boldsymbol{\theta}' - \tilde{\boldsymbol{\theta}}\right\| = \sqrt{4 \cdot 2\epsilon'^2} = 2\sqrt{2}\epsilon' = \epsilon$. Finally,

$$
\|\boldsymbol{\theta}'\|^2 = 4 \cdot \left(\epsilon'^2 + (1 - \epsilon')^2\right) + 4 = 8 + 8\epsilon'(\epsilon' - 1) < 8 = \|\tilde{\boldsymbol{\theta}}\|^2 \;.
$$

### C.6 Proof of Theorem 4.3

We assume w.l.o.g. that the second layer is fully-connected, namely, all hidden neurons are connected to the output neuron, since otherwise we can ignore disconnected neurons. For the network $\Phi$ we use the parameterization $\boldsymbol{\theta} = [\mathbf{w}_1, \dots, \mathbf{w}_k, \mathbf{v}]$ introduced in Section C.1. Thus, we have $\Phi(\boldsymbol{\theta}; \mathbf{x}) = \sum_{l \in [k]} v_l \sigma(\mathbf{w}_l^\top \mathbf{x}^l)$.

We denote $\tilde{\boldsymbol{\theta}} = [\tilde{\mathbf{w}}_1, \ldots, \tilde{\mathbf{w}}_k, \tilde{\mathbf{v}}]$. Since $\tilde{\boldsymbol{\theta}}$ is a KKT point of Problem 2, then there are $\lambda_1, \ldots, \lambda_n$ such that for every $j \in [k]$ we have

$$\tilde{\mathbf{w}}_j = \sum_{i \in [n]} \lambda_i \nabla_{\mathbf{w}_j} \left( y_i \Phi(\tilde{\boldsymbol{\theta}}; \mathbf{x}_i) \right) = \sum_{i \in [n]} \lambda_i y_i \tilde{v}_j \sigma'(\tilde{\mathbf{w}}_j^\top \mathbf{x}_i^j) \mathbf{x}_i^j , \tag{10}$$

and we have $\lambda_i \geq 0$ for all $i$, and $\lambda_i = 0$ if $y_i \Phi(\tilde{\boldsymbol{\theta}}; \mathbf{x}_i) = y_i \sum_{l \in [k]} \tilde{v}_l \sigma(\tilde{\mathbf{w}}_l^\top \mathbf{x}_i^l) \neq 1$. Note that the KKT condition should be w.r.t. the Clarke subdifferential, but since for all $i, j$ we have $\tilde{\mathbf{w}}_j^\top \mathbf{x}_i^j \neq 0$ by our assumption, then we can use here the gradient. By Theorem 2.1, we also have $\lim_{t \to \infty} \|\boldsymbol{\theta}(t)\| = \infty$. Hence, by Lemma C.4 we have $\|\tilde{\mathbf{w}}_j\| = |\tilde{v}_j|$ for all $j \in [k]$.

For $i \in [n]$ and $j \in [k]$ let $A_{ij} = \mathbb{1}(\tilde{\mathbf{w}}_j^\top \mathbf{x}_i^j \geq 0)$. Consider the following problem

$$\min \sum_{l \in [k]} \|\mathbf{u}_l\| \quad \text{s.t.} \quad \forall i \in [n] \; y_i \sum_{l \in [k]} A_{il} \mathbf{u}_l^\top \mathbf{x}_i^l \geq 1 . \tag{11}$$

For every $l \in [k]$ let $\tilde{\mathbf{u}}_l = \tilde{v}_l \cdot \tilde{\mathbf{w}}_l$. Since we assume that the inputs to all neurons in the computations $\Phi(\tilde{\boldsymbol{\theta}}; \mathbf{x}_i)$ are non-zero, then we must have $\tilde{\mathbf{w}}_l \neq \mathbf{0}$ for every $l \in [k]$. Since we also have $\|\tilde{\mathbf{w}}_l\| = |\tilde{v}_l|$, then $\tilde{\mathbf{u}}_l \neq \mathbf{0}$ for all $l \in [k]$. Note that since $\tilde{\mathbf{w}}_1, \ldots, \tilde{\mathbf{w}}_k, \tilde{\mathbf{v}}$ satisfy the constraints in Probelm 2, then $\tilde{\mathbf{u}}_1, \ldots, \tilde{\mathbf{u}}_k$ satisfy the constraints in the above problem. Indeed, for every $i \in [n]$ we have

$$y_i \sum_{l \in [k]} A_{il} \tilde{\mathbf{u}}_l^\top \mathbf{x}_i^l = y_i \sum_{l \in [k]} \mathbb{1}(\tilde{\mathbf{w}}_l^\top \mathbf{x}_i^l \geq 0) \tilde{v}_l \tilde{\mathbf{w}}_l^\top \mathbf{x}_i^l = y_i \sum_{l \in [k]} \tilde{v}_l \sigma(\tilde{\mathbf{w}}_l^\top \mathbf{x}_i^l) \geq 1 .$$

In order to show that $\tilde{\mathbf{u}}_1, \ldots, \tilde{\mathbf{u}}_k$ satisfy the KKT condition of Probelm 11, we need to prove that for every $j \in [k]$ we have

$$\frac{\tilde{\mathbf{u}}_j}{\|\tilde{\mathbf{u}}_j\|} = \sum_{i \in [n]} \lambda_i' y_i A_{ij} \mathbf{x}_i^j \tag{12}$$

for some $\lambda_1', \ldots, \lambda_n'$ such that for all $i$ we have $\lambda_i' \geq 0$, and $\lambda_i' = 0$ if $y_i \sum_{l \in [k]} A_{il} \tilde{\mathbf{u}}_l^\top \mathbf{x}_i^l \neq 1$. From Eq. 10 and since $\|\tilde{\mathbf{w}}_l\| = |\tilde{v}_l|$ for every $l \in [k]$, we have

$$\tilde{\mathbf{u}}_j = \tilde{v}_j \cdot \tilde{\mathbf{w}}_j = \tilde{v}_j \sum_{i \in [n]} \lambda_i y_i \tilde{v}_j A_{ij} \mathbf{x}_i^j = \tilde{v}_j^2 \sum_{i \in [n]} \lambda_i y_i A_{ij} \mathbf{x}_i^j = \|\tilde{v}_j \tilde{\mathbf{w}}_j\| \sum_{i \in [n]} \lambda_i y_i A_{ij} \mathbf{x}_i^j$$

$$= \|\tilde{\mathbf{u}}_j\| \sum_{i \in [n]} \lambda_i y_i A_{ij} \mathbf{x}_i^j .$$

Note that we have $\lambda_i \geq 0$ for all $i$, and $\lambda_i = 0$ if

$$y_i \sum_{l \in [k]} A_{il} \tilde{\mathbf{u}}_l^\top \mathbf{x}_i^l = y_i \sum_{l \in [k]} \tilde{v}_l \mathbb{1}(\tilde{\mathbf{w}}_l^\top \mathbf{x}_i^l \geq 0) \tilde{\mathbf{w}}_l^\top \mathbf{x}_i^l = y_i \sum_{l \in [k]} \tilde{v}_l \sigma(\tilde{\mathbf{w}}_l^\top \mathbf{x}_i^l) \neq 1 .$$

Hence Eq. 12 holds with $\lambda_1', \ldots, \lambda_n'$ that satisfy the requirement. Since the objective in Problem 11 is convex and the constraints are affine functions, then its KKT condition is sufficient for global optimality. Namely, $\tilde{\mathbf{u}}_1, \ldots, \tilde{\mathbf{u}}_k$ are a global optimum for Problem 11.

We now deduce that $\tilde{\boldsymbol{\theta}}$ is a local optimum for Problem 2. Since for every $i \in [n]$ and $l \in [k]$ we have $\tilde{\mathbf{w}}_l^\top \mathbf{x}_i^l \neq 0$, then there is $\epsilon > 0$, such that for every $i, l$ and every $\mathbf{w}_l'$ with $\|\mathbf{w}_l' - \tilde{\mathbf{w}}_l\| \leq \epsilon$ we have $\mathbb{1}(\tilde{\mathbf{w}}_l^\top \mathbf{x}_i^l \geq 0) = \mathbb{1}(\mathbf{w}_l'^\top \mathbf{x}_i^l \geq 0)$. Assume toward contradiction that there is a solution $\boldsymbol{\theta}' = [\mathbf{w}_1', \ldots, \mathbf{w}_k', \mathbf{v}']$ for the constraints in Problem 2 such that $\|\boldsymbol{\theta}' - \tilde{\boldsymbol{\theta}}\| \leq \epsilon$ and $\|\boldsymbol{\theta}'\|^2 < \|\tilde{\boldsymbol{\theta}}\|^2$. Note that we have $\|\mathbf{w}_l' - \tilde{\mathbf{w}}_l\| \leq \epsilon$ for every $l \in [k]$. We denote $\mathbf{u}_l' = v_l' \mathbf{w}_l'$. The vectors $\mathbf{u}_1', \ldots, \mathbf{u}_k'$ satisfy the constraints in Problem 11, since we have

$$y_i \sum_{l \in [k]} A_{il} \mathbf{u}_l'^\top \mathbf{x}_i^l = y_i \sum_{l \in [k]} \mathbb{1}(\tilde{\mathbf{w}}_l^\top \mathbf{x}_i^l \geq 0) v_l' \mathbf{w}_l'^\top \mathbf{x}_i^l = y_i \sum_{l \in [k]} \mathbb{1}(\mathbf{w}_l'^\top \mathbf{x}_i^l \geq 0) v_l' \mathbf{w}_l'^\top \mathbf{x}_i^l$$

$$= y_i \sum_{l \in [k]} v_l' \sigma(\mathbf{w}_l'^\top \mathbf{x}_i^l) \geq 1 ,$$

where the last inequality is since $\boldsymbol{\theta}'$ satisfies the constraints in Probelm 2. Moreover, we have

$$\sum_{l\in[k]}\|\mathbf{u}_l'\| = \sum_{l\in[k]}|v_l'|\cdot\|\mathbf{w}_l'\| \le \sum_{l\in[k]}\frac{1}{2}\left(|v_l'|^2+\|\mathbf{w}_l'\|^2\right) = \frac{1}{2}\|\boldsymbol{\theta}'\|^2 < \frac{1}{2}\left\|\tilde{\boldsymbol{\theta}}\right\|^2$$

$$= \sum_{l\in[k]}\frac{1}{2}\left(|\tilde{v}_l|^2+\|\tilde{\mathbf{w}}_l\|^2\right) .$$

Since $\|\tilde{\mathbf{w}}_l\| = |\tilde{v}_l|$, the above equals

$$\sum_{l\in[k]}\|\tilde{\mathbf{w}}_l\|^2 = \sum_{l\in[k]}|\tilde{v}_l|\cdot\|\tilde{\mathbf{w}}_l\| = \sum_{l\in[k]}\|\tilde{\mathbf{u}}_l\| ,$$

which contradicts the global optimality of $\tilde{\mathbf{u}}_1,\dots,\tilde{\mathbf{u}}_k$.

It remains to show that $\tilde{\boldsymbol{\theta}}$ may not be a global optimum of Problem 2, even if the network $\Phi$ is fully connected. The following lemma concludes the proof.

**Lemma C.6.** *Let $\Phi$ be a depth-2 fully-connected ReLU network with input dimension 2 and two hidden neurons. Consider minimizing either the exponential or the logistic loss using GF. Then, there exists a dataset $\{(\mathbf{x}_i,y_i)\}_{i=1}^n$ and an initialization $\boldsymbol{\theta}(0)$, such that GF converges to zero loss, converges in direction to a KKT point $\tilde{\boldsymbol{\theta}} = [\tilde{\mathbf{w}}_1,\tilde{\mathbf{w}}_2,\tilde{\mathbf{v}}]$ of Problem 2 such that $\langle\tilde{\mathbf{w}}_j,\mathbf{x}_i\rangle \ne 0$ for all $j \in \{1,2\}$ and $i \in [n]$, and $\tilde{\boldsymbol{\theta}}$ is not a global optimum.*

*Proof.* Let $\mathbf{x}_1 = \left(1,\frac{1}{4}\right)^\top$, $\mathbf{x}_2 = \left(-1,\frac{1}{4}\right)^\top$, $\mathbf{x}_3 = (0,-1)$, $y_1 = y_2 = y_3 = 1$. Let $\{(\mathbf{x}_1,y_1),(\mathbf{x}_2,y_2),(\mathbf{x}_3,y_3)\}$ be a dataset. Consider the initialization $\boldsymbol{\theta}(0)$ such that $\mathbf{w}_1(0) = (0,3)$, $v_1(0) = 3$, $\mathbf{w}_2(0) = (0,-2)$ and $v_2(0) = 2$. Note that $\mathcal{L}(\boldsymbol{\theta}(0)) = 2\ell\left(\frac{9}{4}\right)+\ell(4) < 1$ for both the exponential loss and the logistic loss, and therefore by Theorem 2.1 GF converges in direction to a KKT point $\tilde{\boldsymbol{\theta}}$ of Problem 2.

Note that for $\boldsymbol{\theta}$ such that $\mathbf{w}_1 = \alpha\cdot(0,1)^\top$ and $\mathbf{w}_2 = \beta\cdot(0,-1)^\top$ for some $\alpha,\beta > 0$, and $v_1,v_2 > 0$, we have

$$\nabla_{\mathbf{w}_1}\mathcal{L}(\boldsymbol{\theta}) = \sum_{i=1}^3\ell'(y_i\Phi(\boldsymbol{\theta};\mathbf{x}_i))\cdot y_i\nabla_{\mathbf{w}_1}\Phi(\boldsymbol{\theta};\mathbf{x}_i)$$

$$= \sum_{i=1}^3\ell'\left(v_1\sigma(\mathbf{w}_1^\top\mathbf{x}_i)+v_2\sigma(\mathbf{w}_2^\top\mathbf{x}_i)\right)\cdot v_1\sigma'(\mathbf{w}_1^\top\mathbf{x}_i)\mathbf{x}_i$$

$$= \sum_{i=1}^2\ell'(v_1\sigma(\mathbf{w}_1^\top\mathbf{x}_i))\cdot v_1\mathbf{x}_i = v_1\ell'\left(v_1\frac{\alpha}{4}\right)\sum_{i=1}^2\mathbf{x}_i ,$$

and

$$\nabla_{v_1}\mathcal{L}(\boldsymbol{\theta}) = \sum_{i=1}^3\ell'(y_i\Phi(\boldsymbol{\theta};\mathbf{x}_i))\cdot y_i\nabla_{v_1}\Phi(\boldsymbol{\theta};\mathbf{x}_i) = \sum_{i=1}^3\ell'(y_i\Phi(\boldsymbol{\theta};\mathbf{x}_i))\cdot\sigma(\mathbf{w}_1^\top\mathbf{x}_i) .$$

Hence, $-\nabla_{\mathbf{w}_1}\mathcal{L}(\boldsymbol{\theta})$ points in the direction $(0,1)^\top$ and $-\nabla_{v_1}\mathcal{L}(\boldsymbol{\theta}) > 0$. Moreover, we have

$$\nabla_{\mathbf{w}_2}\mathcal{L}(\boldsymbol{\theta}) = \sum_{i=1}^3\ell'(y_i\Phi(\boldsymbol{\theta};\mathbf{x}_i))\cdot y_i\nabla_{\mathbf{w}_2}\Phi(\boldsymbol{\theta};\mathbf{x}_i)$$

$$= \sum_{i=1}^3\ell'\left(v_1\sigma(\mathbf{w}_1^\top\mathbf{x}_i)+v_2\sigma(\mathbf{w}_2^\top\mathbf{x}_i)\right)\cdot v_2\sigma'(\mathbf{w}_2^\top\mathbf{x}_i)\mathbf{x}_i$$

$$= \ell'(v_2\sigma(\mathbf{w}_2^\top\mathbf{x}_3))\cdot v_2\mathbf{x}_3 = v_2\ell'(v_2\beta)\mathbf{x}_3 ,$$

and

$$\nabla_{v_2}\mathcal{L}(\boldsymbol{\theta}) = \sum_{i=1}^3\ell'(y_i\Phi(\boldsymbol{\theta};\mathbf{x}_i))\cdot y_i\nabla_{v_2}\Phi(\boldsymbol{\theta};\mathbf{x}_i) = \sum_{i=1}^3\ell'(v_1\sigma(\mathbf{w}_1^\top\mathbf{x}_i)+v_2\sigma(\mathbf{w}_2^\top\mathbf{x}_i))\cdot\sigma(\mathbf{w}_2^\top\mathbf{x}_i)$$

$$= \ell'(v_2\sigma(\mathbf{w}_2^\top\mathbf{x}_3))\sigma(\mathbf{w}_2^\top\mathbf{x}_3) = \ell'(v_2\beta)\cdot\beta .$$

Therefore, $-\nabla_{\mathbf{w}_2}\mathcal{L}(\boldsymbol{\theta})$ points in the direction $(0,-1)^\top$ and $-\nabla_{v_2}\mathcal{L}(\boldsymbol{\theta}) > 0$. Hence for every $t$ we have $\mathbf{w}_1(t) = \alpha(t) \cdot (0,1)^\top$ for some $\alpha(t) > 0$ and $v_1(t) > 0$. Also, we have $\mathbf{w}_2(t) = \beta(t) \cdot (0,-1)^\top$ for some $\beta(t) > 0$ and $v_2(t) > 0$. By Lemma C.1, we have for every $t \geq 0$ that $\|\mathbf{w}_1(t)\|^2 - v_1(t)^2 = \|\mathbf{w}_1(0)\|^2 - v_1(0)^2 = 0$ and $\|\mathbf{w}_2(t)\|^2 - v_2(t)^2 = \|\mathbf{w}_2(0)\|^2 - v_2(0)^2 = 0$. Hence, we have $v_1(t) = \alpha(t)$ and $v_2(t) = \beta(t)$. Therefore, we have $\tilde{\mathbf{w}}_1 = \tilde{\alpha} \cdot (0,1)^\top$ and $\tilde{v}_1 = \tilde{\alpha}$ for some $\tilde{\alpha} \geq 0$. Likewise, we have $\tilde{\mathbf{w}}_2 = \tilde{\beta} \cdot (0,-1)^\top$ and $\tilde{v}_2 = \tilde{\beta}$ for some $\tilde{\beta} \geq 0$. Since $\tilde{\boldsymbol{\theta}}$ satisfies the constraints in Probelm 2, then $\tilde{\alpha} \geq 2$ and $\tilde{\beta} \geq 1$. Note that $\langle \tilde{\mathbf{w}}_j, \mathbf{x}_i \rangle \neq 0$ for all $j \in \{1,2\}$ and $i \in \{1,2,3\}$.

We now show that there exists a solution $\boldsymbol{\theta}'$ to Problem 2 with a smaller norm, and hence $\tilde{\boldsymbol{\theta}}$ is not a global optimum. Let $\boldsymbol{\theta}' = [\mathbf{w}'_1, \mathbf{w}'_2, \mathbf{v}']$ such that $\mathbf{w}'_1 = \frac{\mathbf{x}_1}{\tilde{\alpha}\|\mathbf{x}_1\|}$, $v'_1 = \tilde{\alpha}$, $\mathbf{w}'_2 = \frac{1}{\tilde{\beta}} \cdot \left(-\frac{5}{4}, -1\right)$, and $v'_2 = \tilde{\beta}$. It is easy to verify that $\boldsymbol{\theta}'$ satisfies the constraints in Problem 2, and we have

$$\|\boldsymbol{\theta}'\|^2 = \frac{1}{\tilde{\alpha}^2} + \tilde{\alpha}^2 + \frac{1}{\tilde{\beta}^2}\left(\frac{25}{16} + 1\right) + \tilde{\beta}^2 < \frac{1}{4} + \tilde{\alpha}^2 + 1 \cdot 3 + \tilde{\beta}^2 < \tilde{\beta}^2 + \tilde{\alpha}^2 + \tilde{\alpha}^2 + \tilde{\beta}^2 = \|\tilde{\boldsymbol{\theta}}\|^2 .$$

$\square$

## C.7 Proof of Theorem 4.4

Let $\mathbf{x} = \left(4, \frac{1}{\sqrt{2}}, -4, \frac{1}{\sqrt{2}}\right)^\top$ and $y = 1$. Let $\boldsymbol{\theta}(0) = [\mathbf{w}(0), \mathbf{v}(0)]$ where $\mathbf{w}(0) = (0,1)^\top$ and $\mathbf{v}(0) = \left(\frac{1}{\sqrt{2}}, \frac{1}{\sqrt{2}}\right)^\top$. Note that $\Psi(\boldsymbol{\theta}(0); \mathbf{x}) = 1$ and hence $\mathcal{L}(\boldsymbol{\theta}(0)) < 1$ for both the exponential loss and the logistic loss. Therefore, by Theorem 2.1 GF converges in direction to a KKT point $\tilde{\boldsymbol{\theta}}$ of Problem 2, and we have $\lim_{t\to\infty}\mathcal{L}(\boldsymbol{\theta}(t)) = 0$ and $\lim_{t\to\infty}\|\boldsymbol{\theta}(t)\| = \infty$.

The symmetry of the input $\mathbf{x}$ and the initialization $\boldsymbol{\theta}(0)$ implies that the direction of $\mathbf{w}$ does not change during the training, and that we have $v_1(t) = v_2(t) > 0$ for all $t \geq 0$. More formally, this claim follows from the following calculation. For $j \in \{1,2\}$ we have

$$\nabla_{v_j}\mathcal{L}(\boldsymbol{\theta}) = \ell'(y\Phi(\boldsymbol{\theta};\mathbf{x})) \cdot y\nabla_{v_j}\Phi(\boldsymbol{\theta};\mathbf{x}) = \ell'(y\Phi(\boldsymbol{\theta};\mathbf{x})) \cdot \sigma(\mathbf{w}^\top\mathbf{x}^{(j)}) .$$

Moreover,

$$\nabla_{\mathbf{w}}\mathcal{L}(\boldsymbol{\theta}) = \ell'(y\Phi(\boldsymbol{\theta};\mathbf{x})) \cdot y\nabla_{\mathbf{w}}\Phi(\boldsymbol{\theta};\mathbf{x}) = \ell'(y\Phi(\boldsymbol{\theta};\mathbf{x})) \cdot \left(v_1\sigma'(\mathbf{w}^\top\mathbf{x}^{(1)})\mathbf{x}^{(1)} + v_2\sigma'(\mathbf{w}^\top\mathbf{x}^{(2)})\mathbf{x}^{(2)}\right) .$$

Hence, if $v_1 = v_2 > 0$ and $\mathbf{w}$ points in the direction $(0,1)^\top$, then it is easy to verify that $\nabla_{v_1}\mathcal{L}(\boldsymbol{\theta}) = \nabla_{v_1}\mathcal{L}(\boldsymbol{\theta}) < 0$ and that $\nabla_{\mathbf{w}}\mathcal{L}(\boldsymbol{\theta})$ points in the direction of $-(\mathbf{x}^{(1)}+\mathbf{x}^{(2)}) = -(0, \sqrt{2})^\top$. Furthermore, by Lemma C.2, for every $t \geq 0$ we have $\|\mathbf{w}(t)\|^2 - \|\mathbf{v}(t)\|^2 = \|\mathbf{w}(0)\|^2 - \|\mathbf{v}(0)\|^2 = 0$.

Therefore, the KKT point $\tilde{\boldsymbol{\theta}} = [\tilde{\mathbf{w}}, \tilde{\mathbf{v}}]$ is such that $\tilde{\mathbf{w}}$ points at the direction $(0,1)^\top$, $\tilde{v}_1 = \tilde{v}_2 > 0$, and $\|\tilde{\mathbf{w}}\| = \|\tilde{\mathbf{v}}\|$. Since $\tilde{\boldsymbol{\theta}}$ satisfies the KKT conditions of Problem 2, then we have

$$\tilde{\mathbf{w}} = \lambda\nabla_{\mathbf{w}}\left(y\Phi(\tilde{\boldsymbol{\theta}};\mathbf{x})\right) ,$$

where $\lambda \geq 0$ and $\lambda = 0$ if $y\Phi(\tilde{\boldsymbol{\theta}};\mathbf{x}) \neq 1$. Hence, we must have $y\Phi(\tilde{\boldsymbol{\theta}};\mathbf{x}) = 1$. Letting $z := \tilde{v}_1 = \tilde{v}_2$ and using $2z^2 = \|\tilde{\mathbf{v}}\|^2 = \|\tilde{\mathbf{w}}\|^2 = \tilde{w}_2^2$, we have

$$1 = \tilde{v}_1\sigma(\tilde{\mathbf{w}}^\top\mathbf{x}^{(1)}) + \tilde{v}_2\sigma(\tilde{\mathbf{w}}^\top\mathbf{x}^{(2)}) = z\tilde{\mathbf{w}}^\top\mathbf{x}^{(1)} + z\tilde{\mathbf{w}}^\top\mathbf{x}^{(2)} = z\tilde{\mathbf{w}}^\top\left(\mathbf{x}^{(1)} + \mathbf{x}^{(2)}\right) = z \cdot \tilde{w}_2\sqrt{2}$$

$$= \frac{\tilde{w}_2}{\sqrt{2}} \cdot \tilde{w}_2\sqrt{2} = \tilde{w}_2^2 .$$

Therefore, $\tilde{\mathbf{w}} = (0,1)^\top$ and $\tilde{\mathbf{v}} = \left(\frac{1}{\sqrt{2}}, \frac{1}{\sqrt{2}}\right)$. Note that we have $\langle\tilde{\mathbf{w}}, \mathbf{x}^{(1)}\rangle \neq 0$ and $\langle\tilde{\mathbf{w}}, \mathbf{x}^{(2)}\rangle \neq 0$.

It remains to show that $\tilde{\boldsymbol{\theta}}$ is not a local optimum of Problem 2. We show that for every $0 < \epsilon' < 1$ there exists some $\boldsymbol{\theta}' = [\mathbf{w}', \mathbf{v}']$ such that $\left\|\boldsymbol{\theta}' - \tilde{\boldsymbol{\theta}}\right\| \leq \epsilon'$, $\boldsymbol{\theta}'$ satisfies the constrains in Problem 2, and

$\|\boldsymbol{\theta}'\| < \|\tilde{\boldsymbol{\theta}}\|$. Let $\epsilon = \frac{\epsilon'^2}{2} \in (0, 1/2)$, and let $\mathbf{w}' = (\sqrt{\epsilon}, 1 - \epsilon)^\top$ and $\mathbf{v}' = \left(\frac{1}{\sqrt{2}} + \frac{\sqrt{\epsilon}}{2}, \frac{1}{\sqrt{2}} - \frac{\sqrt{\epsilon}}{2}\right)^\top$. Note that

$$\|\boldsymbol{\theta}'\|^2 = \|\mathbf{w}'\|^2 + \|\mathbf{v}'\|^2 = \epsilon + (1 - \epsilon)^2 + \left(\frac{1}{\sqrt{2}} + \frac{\sqrt{\epsilon}}{2}\right)^2 + \left(\frac{1}{\sqrt{2}} - \frac{\sqrt{\epsilon}}{2}\right)^2$$

$$= \epsilon + 1 + \epsilon^2 - 2\epsilon + 1 + \frac{\epsilon}{2} = 2 - \frac{\epsilon}{2} + \epsilon^2 < 2 - \frac{\epsilon}{2} + \frac{\epsilon}{2} = 2 = \|\tilde{\mathbf{w}}\|^2 + \|\tilde{\mathbf{v}}\|^2 = \left\|\tilde{\boldsymbol{\theta}}\right\|^2 .$$

Moreover,

$$\left\|\boldsymbol{\theta}' - \tilde{\boldsymbol{\theta}}\right\|^2 = \epsilon + \epsilon^2 + \frac{\epsilon}{4} + \frac{\epsilon}{4} = \epsilon^2 + \frac{3\epsilon}{2} = \frac{\epsilon'^4}{4} + \frac{3\epsilon'^2}{4} < \frac{\epsilon'^2}{4} + \frac{3\epsilon'^2}{4} = \epsilon'^2 .$$

Finally, we show that $\boldsymbol{\theta}'$ satisfies the constraints:

$$\Phi(\boldsymbol{\theta}'; \mathbf{x}) = v_1'\sigma(\mathbf{w}'^\top \mathbf{x}^{(1)}) + v_2'\sigma(\mathbf{w}'^\top \mathbf{x}^{(2)})$$

$$= \left(\frac{1}{\sqrt{2}} + \frac{\sqrt{\epsilon}}{2}\right)\left(4\sqrt{\epsilon} + \frac{1}{\sqrt{2}} \cdot (1 - \epsilon)\right) + \left(\frac{1}{\sqrt{2}} - \frac{\sqrt{\epsilon}}{2}\right)\left(-4\sqrt{\epsilon} + \frac{1}{\sqrt{2}} \cdot (1 - \epsilon)\right)$$

$$= \frac{1}{\sqrt{2}} \cdot (1 - \epsilon)\left(\frac{1}{\sqrt{2}} + \frac{\sqrt{\epsilon}}{2} + \frac{1}{\sqrt{2}} - \frac{\sqrt{\epsilon}}{2}\right) + 4\sqrt{\epsilon}\left(\frac{1}{\sqrt{2}} + \frac{\sqrt{\epsilon}}{2} - \frac{1}{\sqrt{2}} + \frac{\sqrt{\epsilon}}{2}\right)$$

$$= 1 - \epsilon + 4\epsilon = 1 + 3\epsilon \geq 1 .$$

### C.8 Proof of Theorem 5.1

Let $\mathbf{x} = (1, 1)^\top$ and $y = 1$. Consider the initialization $\boldsymbol{\theta}(0) = [\mathbf{w}_1(0), \ldots, \mathbf{w}_m(0)]$, where $\mathbf{w}_j(0) = (1, 1)^\top$ for every $j \in [m]$. Note that $\mathcal{L}(\boldsymbol{\theta}(0)) = \ell(2) < 1$ for both linear and ReLU networks with the exponential loss or the logistic loss, and therefore by Theorem 2.1 GF converges in direction to a KKT point $\tilde{\boldsymbol{\theta}}$ of Problem 2, and we have $\lim_{t\to\infty} \mathcal{L}(\boldsymbol{\theta}(t)) = 0$ and $\lim_{t\to\infty} \|\boldsymbol{\theta}(t)\| = \infty$. It remains to show that it does not converge in direction to a local optimum of Problem 2.

From the symmetry of the network $\Phi$ and the initialization $\boldsymbol{\theta}(0)$, it follows that for all $t$ the network $\Phi(\boldsymbol{\theta}(t); \cdot)$ remains symmetric, namely, there are $\alpha_j(t)$ such that $\mathbf{w}_j(t) = (\alpha_j(t), \alpha_j(t))$. Moreover, by Lemma C.2, for every $t \geq 0$ and $j, l \in [m]$ we have $\alpha_j(t) = \alpha_l(t) := \alpha(t)$. Thus, GF converges in direction to the KKT point $\tilde{\boldsymbol{\theta}} = [\tilde{\mathbf{w}}_1, \ldots, \tilde{\mathbf{w}}_m]$ such that $\tilde{\mathbf{w}}_j = \left(2^{-1/m}, 2^{-1/m}\right)^\top$ for all $j \in [m]$. Note that since the dataset is of size 1, then every KKT point of Problem 2 must label the input $\mathbf{x}$ with exactly 1.

We now show that $\tilde{\boldsymbol{\theta}}$ is not a local optimum of Problem 2. The following arguments hold for both linear and ReLU networks. Let $0 < \epsilon < \frac{1}{2}$. Let $\boldsymbol{\theta}' = [\mathbf{w}_1', \ldots, \mathbf{w}_m']$ such that for every $j \in [m]$ we have $\mathbf{w}_j' = \left(\left(\frac{1+\epsilon}{2}\right)^{1/m}, \left(\frac{1-\epsilon}{2}\right)^{1/m}\right)^\top$. We have

$$y \cdot \Phi(\boldsymbol{\theta}'; \mathbf{x}) = \left(\frac{1+\epsilon}{2}\right) + \left(\frac{1-\epsilon}{2}\right) = 1 .$$

Hence, $\boldsymbol{\theta}'$ satisfies the constraints in Problem 2. We now show that for every sufficiently small $\epsilon > 0$ we have $\|\boldsymbol{\theta}'\|^2 < \|\tilde{\boldsymbol{\theta}}\|^2$. We need to show that

$$m\left(\frac{1+\epsilon}{2}\right)^{2/m} + m\left(\frac{1-\epsilon}{2}\right)^{2/m} < 2m\left(\frac{1}{2}\right)^{2/m} .$$

Therefore, it suffices to show that

$$(1+\epsilon)^{2/m} + (1-\epsilon)^{2/m} < 2 .$$

Let $g : \mathbb{R} \to \mathbb{R}$ such that $g(s) = (1+s)^{2/m} + (1-s)^{2/m}$. We have $g(0) = 2$. The derivatives of $g$ satisfy

$$g'(s) = \frac{2}{m}(1+s)^{\frac{2}{m}-1} - \frac{2}{m}(1-s)^{\frac{2}{m}-1} ,$$

and

$$g''(s) = \frac{2}{m}\left(\frac{2}{m} - 1\right)(1 + s)^{\frac{2}{m} - 2} + \frac{2}{m}\left(\frac{2}{m} - 1\right)(1 - s)^{\frac{2}{m} - 2} \ .$$

Since $m \geq 3$ we have $g'(0) = 0$ and $g''(0) < 0$. Hence, $0$ is a local maximum of $g$. Therefore for every sufficiently small $\epsilon > 0$ we have $g(\epsilon) < 2$ and thus $\|\boldsymbol{\theta}'\|^2 < \|\tilde{\boldsymbol{\theta}}\|^2$.

Finally, note that the inputs to all neurons in the computation $\Phi(\tilde{\boldsymbol{\theta}}; \mathbf{x})$ are positive.

## C.9 Proof of Theorem 5.2

By Theorem 2.1 GF converge in direction to a KKT point $\tilde{\boldsymbol{\theta}} = [\tilde{\mathbf{u}}^{(l)}]_{l=1}^{m}$ of Problem 2. We now show that for every layer $l \in [m]$ the parameters vector $\tilde{\mathbf{u}}^{(l)}$ is a global optimum of Problem 4 w.r.t. $\tilde{\boldsymbol{\theta}}$.

Since $\tilde{\boldsymbol{\theta}}$ is a KKT point of Problem 2, then there are $\lambda_1, \ldots, \lambda_n$ such that for every $l \in [m]$ we have

$$\tilde{\mathbf{u}}^{(l)} = \sum_{i \in [n]} \lambda_i \frac{\partial\left(y_i \Phi(\tilde{\boldsymbol{\theta}}; \mathbf{x}_i)\right)}{\partial \mathbf{u}^{(l)}} \ ,$$

where $\lambda_i \geq 0$ for all $i$, and $\lambda_i = 0$ if $y_i \Phi(\tilde{\boldsymbol{\theta}}; \mathbf{x}_i) \neq 1$. Letting $\boldsymbol{\theta}'(\mathbf{u}^{(l)}) = [\tilde{\mathbf{u}}^{(1)}, \ldots, \tilde{\mathbf{u}}^{(l-1)}, \mathbf{u}^{(l)}, \tilde{\mathbf{u}}^{(l+1)}, \ldots, \tilde{\mathbf{u}}^{(m)}]$, the above equation can be written as

$$\tilde{\mathbf{u}}^{(l)} = \sum_{i \in [n]} \lambda_i \frac{\partial\left(y_i \Phi(\boldsymbol{\theta}'(\tilde{\mathbf{u}}^{(l)}); \mathbf{x}_i)\right)}{\partial \mathbf{u}^{(l)}} \ ,$$

where $\lambda_i \geq 0$ for all $i$, and $\lambda_i = 0$ if $y_i \Phi(\boldsymbol{\theta}'(\tilde{\mathbf{u}}^{(l)}); \mathbf{x}_i) = y_i \Phi(\tilde{\boldsymbol{\theta}}; \mathbf{x}_i) \neq 1$. Moreover, if the constraints in Problem 2 are satisfies in $\tilde{\boldsymbol{\theta}}$, then the constrains in Problem 4 are also satisfied for every $l \in [m]$ in $\tilde{\mathbf{u}}^{(l)}$ w.r.t. $\tilde{\boldsymbol{\theta}}$. Hence, for every $l \in [m]$ the KKT conditions of Problem 4 w.r.t. $\tilde{\boldsymbol{\theta}}$ hold. Since the constraints in Problem 4 are affine and the objective is convex, then this KKT point is a global optimum.

## C.10 Proof of Theorem 5.3

Let $\{(\mathbf{x}_i, y_i)\}_{i=1}^{4}$ be a dataset such that $y_i = 1$ for all $i \in [4]$ and we have $\mathbf{x}_1 = (0, 1)^\top$, $\mathbf{x}_2 = (1, 0)^\top$, $\mathbf{x}_3 = (0, -1)$ and $\mathbf{x}_4 = (-1, 0)$. In the proof of Theorem 4.2 (part 2) we showed that for an appropriate initialization, for both the exponential loss and the logistic loss GF converges to zero loss, and converges in direction to a KKT point $\tilde{\boldsymbol{\theta}}$ of Problem 2. Moreover, in the proof of Theorem 4.2 we showed that the KKT point $\tilde{\boldsymbol{\theta}}$ is such that for all $j \in [4]$ we have $\tilde{\mathbf{w}}_j = \mathbf{x}_j$ and $\tilde{v}_j = 1$.

We show that $\tilde{\mathbf{w}}_1, \tilde{\mathbf{w}}_2, \tilde{\mathbf{w}}_3, \tilde{\mathbf{w}}_4$ is not a local optimum of Problem 4 w.r.t. $\tilde{\boldsymbol{\theta}}$. It suffices to prove that for every $0 < \epsilon < 1$ there exists some $\boldsymbol{\theta}'$ such that $v'_j = \tilde{v}_j$ for all $j \in [4]$, $\|\boldsymbol{\theta}' - \tilde{\boldsymbol{\theta}}\| \leq \epsilon$, $\boldsymbol{\theta}'$ satisfies the constraints, and $\|\boldsymbol{\theta}'\| < \|\tilde{\boldsymbol{\theta}}\|$. The existence of such $\boldsymbol{\theta}'$ is shown in the proof of Theorem 4.2. Hence, we conclude the proof of the theorem.

## C.11 Proof of Theorem 5.4

By Theorem 2.1 GF converge in direction to a KKT point $\tilde{\boldsymbol{\theta}} = [\tilde{\mathbf{u}}^{(l)}]_{l=1}^{m}$ of Problem 2. Let $l \in [m]$ and assume that for every $i \in [n]$ the inputs to all neurons in layers $l, \ldots, m - 1$ in the computation $\Phi(\tilde{\boldsymbol{\theta}}; \mathbf{x}_i)$ are non-zero. We now show that the parameters vector $\tilde{\mathbf{u}}^{(l)}$ is a local optimum of Problem 4 w.r.t. $\tilde{\boldsymbol{\theta}}$.

For $i \in [n]$ and $k \in [m - 1]$ we denote by $\mathbf{x}_i^{(k)} \in \mathbb{R}^{d_k}$ the output of the $k$-th layer in the computation $\Phi(\tilde{\boldsymbol{\theta}}; \mathbf{x}_i)$, and denote $\mathbf{x}_i^{(0)} = \mathbf{x}_i$. If $l \in [m - 1]$ then we define the following notations. We denote by $f_l : \mathbb{R}^{d_l} \to \mathbb{R}$ the function computed by layers $l + 1, \ldots, m$ of $\Phi(\tilde{\boldsymbol{\theta}}; \cdot)$. Thus, we have $\Phi(\tilde{\boldsymbol{\theta}}; \mathbf{x}_i) = f_l(\mathbf{x}_i^{(l)}) = f_l \circ \sigma\left(\tilde{W}^{(l)} \mathbf{x}_i^{(l-1)}\right)$, where $\tilde{W}^{(l)}$ is the weight matrix that corresponds to $\tilde{\mathbf{u}}^{(l)}$. For $i \in [n]$ we denote by $h_i$ the function $\mathbf{u}^{(l)} \mapsto f_l \circ \sigma(W^{(l)} \mathbf{x}_i^{(l-1)})$ where $W^{(l)}$ is the weights

matrix that corresponds to $\mathbf{u}^{(l)}$. Thus, $\Phi(\tilde{\boldsymbol{\theta}}; \mathbf{x}_i) = h_i(\tilde{\mathbf{u}}^{(l)})$. If $l = m$ then we denote by $h_i$ the function $\mathbf{u}^{(m)} \mapsto W^{(m)} \mathbf{x}_i^{(m-1)}$, thus we also have $\Phi(\tilde{\boldsymbol{\theta}}; \mathbf{x}_i) = h_i(\tilde{\mathbf{u}}^{(m)})$.

Since $\tilde{\boldsymbol{\theta}}$ is a KKT point of Problem 2, then there are $\lambda_1, \dots, \lambda_n$ such that

$$\tilde{\mathbf{u}}^{(l)} = \sum_{i \in [n]} \lambda_i \frac{\partial \left( y_i \Phi(\tilde{\boldsymbol{\theta}}; \mathbf{x}_i) \right)}{\partial \mathbf{u}^{(l)}} = \sum_{i \in [n]} \lambda_i \frac{\partial}{\partial \mathbf{u}^{(l)}} \left[ y_i \cdot h_i(\tilde{\mathbf{u}}^{(l)}) \right] \; ,$$

where $\lambda_i \geq 0$ for all $i$, and $\lambda_i = 0$ if $y_i \cdot h_i(\tilde{\mathbf{u}}^{(l)}) \neq 1$. Note that since the inputs to all neurons in layers $l, \dots, m-1$ in the computation $\Phi(\tilde{\boldsymbol{\theta}}; \mathbf{x}_i)$ are non-zero, then the function $h_i$ is differentiable at $\tilde{\mathbf{u}}^{(l)}$. Therefore in the above KKT condition we use the derivative rather than the Clarke subdifferential. Moreover, if the constraints in Problem 2 are satisfies in $\tilde{\boldsymbol{\theta}}$, then the constrains in Problem 4 are also satisfied in $\tilde{\mathbf{u}}^{(l)}$ w.r.t. $\tilde{\boldsymbol{\theta}}$. Hence, the KKT condition of Problem 4 w.r.t. $\tilde{\boldsymbol{\theta}}$ holds.

Also, note that since the inputs to all neurons in layers $l, \dots, m-1$ in the computation $\Phi(\tilde{\boldsymbol{\theta}}; \mathbf{x}_i)$ are non-zero, then the function $h_i$ is locally linear near $\tilde{\mathbf{u}}^{(l)}$. We denote this linear function by $\tilde{h}_i$. Therefore, $\tilde{\mathbf{u}}^{(l)}$ is a KKT point of the following problem

$$\min_{\mathbf{u}^{(l)}} \frac{1}{2} \left\| \mathbf{u}^{(l)} \right\|^2 \quad \text{s.t.} \quad \forall i \in [n] \quad y_i \tilde{h}_i(\mathbf{u}^{(l)}) \geq 1 \; .$$

Since the constrains here are affine and the objective is convex, then $\tilde{\mathbf{u}}^{(l)}$ is a global optimum of the above problem. Thus, there is a small ball near $\tilde{\mathbf{u}}^{(l)}$ where $\tilde{\mathbf{u}}^{(l)}$ is the optimum of Problem 4 w.r.t. $\tilde{\boldsymbol{\theta}}$, namely, it is a local optimum.

Finally, note that in the proof of Lemma C.6 the parameters vector $\boldsymbol{\theta}'$ is obtained from $\tilde{\boldsymbol{\theta}}$ by changing only the first layer. Hence, in ReLU networks GF might converge in direction to a KKT point of Problem 2 which is not a global optimum of Problem 4, even if all inputs to neurons are non-zero.