# OpenReview forum: "On Margin Maximization in Linear and ReLU Networks"
_NeurIPS.cc/2022/Conference — NeurIPS 2022 Accept_

### Official Review · Reviewer_Uo2f · 2022-06-30

**Rating:** 5
**Confidence:** 4
**Soundness:** 3 good
**Presentation:** 3 good
**Contribution:** 2 fair

**Summary:**

This paper analyzes margin maximization properties of gradient flow (GF) on linear and ReLU neural networks. It was shown by previous work that GF converges to a KKT point of a margin maximization problem when trained on linear or ReLU networks. This paper first considers fully-connected networks, and shows that for linear networks, GF converges to a global optimum of the margin maximization problem, while for ReLU networks, the GF solution may not even be locally optimal. Next, if the network may contain sparse connections (e.g., diagonal networks), it is shown that the GF solution may not be locally optimal even for depth-2 linear networks. If we further require all neurons have nonzero incoming weight vectors in the KKT solution, then it will be globally optimal for depth-2 linear networks, but this condition is still not enough to guarantee local optimality for depth-2 ReLU networks or deep linear networks with depth larger than 2. For depth-2 ReLU networks with sparse connections, if it is further required that the inputs to all hidden neurons are nonzero, then it is shown that GF converges to a local optimum, but it might not be globally optimal.

**Questions:**

N/A

**Limitations:**

As mentioned above, it will he helpful to have a discussion section.

**Strengths And Weaknesses:**

This paper gives a detailed analysis of margin maximization for GF. It is shown that a KKT point may not be globally or even locally optimal in many different settings. On the other hand, with some additional regularity conditions (e.g., no zero input weight vectors), local or global optimality can be guaranteed. I think these results are useful to the community.

On the other hand, it seems many of the negative examples can be handled by having a randomly-initialized network with a reasonable width, which is more relevant to practice. It would be very interesting if there exists a setting that can also fail a wide randomly-initialized network. Additionally, the proof idea of Theorem 3.1 is similar to that of [Ji and Telgarsky 2018, Gradient descent aligns the layers of deep linear networks]; the difference should be discussed.

Finally, the presentation is a little intense, especially at the end of the paper. It will also be helpful to have a section discussing limitations, open problems, etc.

---

> ### Author Response · Authors · 2022-07-31
> **Response**
>
> Thanks for your comments!
>
> “it seems many of the negative examples can be handled by having a randomly-initialized network with a reasonable width…”:
> The negative results help understanding which additional assumptions are required for obtaining positive results. We showed some positive results under certain assumptions, and we agree that establishing more positive results under other assumptions is an intriguing direction for future research.
>
> “the proof idea of Theorem 3.1 …”:
> Our proof of Theorem 3.1 uses Proposition 4.4 from Ji and Telgarsky 2020, which generalizes the result from Ji and Telgarsky 2018. Thus, we do not repeat the work done there.
>
> “the presentation is a little intense… It will also be helpful to have a section discussing limitations, open problems, etc.”:
> We agree that the paper is a little intense. We will try to squeeze in a discussion section.

---

### Official Review · Reviewer_rHbR · 2022-07-03

**Rating:** 6
**Confidence:** 4
**Soundness:** 3 good
**Presentation:** 3 good
**Contribution:** 3 good

**Summary:**

The paper studies margin maximization for linear and ReLU networks with exponential and logistic loss. Under this setting, several prior works could show that gradient flow converges to a KKT point of the max margin problem, but these works did not discuss when the KKT point is a local/global optimum of the max margin problem. The current paper studies this question. For the various settings of the network (linear/ReLU, fully connected/convolutional/weight sharing) and the training data, they show that the KKT solutions are *not* even local optimum of the max margin problem. However, in some special cases such as deep linear nets with fully connected architecture, they can show the global optimality.
The main results of the paper can be summarized as follows.

- Global optimality for deep linear nets with dense connections: every KKT point is a global optimum of the max margin problem. This result also holds when the network has sparse connections under an additional assumption that the KKT point has non-zero incoming weights.

- For 2-layer ReLU nets: KKT points with non-zero inputs to hidden neurons are local optimum, but they are not necessarily global. The non-zero input assumption appears to be important as they show there exists a dataset where the same network with 2 inputs and 2 hidden neurons does not have this property.

- For deep ReLU nets: KKT points with non-zero inputs to all neurons satisfy a per-layer local optimality. As before, the non-zero input assumptions seems to be necessary as they show there exists a dataset where a 2-layer ReLU net with 2 inputs and 4 hidden neurons may not have this property. However, for deep linear nets, per-layer global optimality always holds.

**Questions:**

See Weakness.

**Limitations:**

It would be nice to improve the generality of the counterexamples in the paper, e.g. by replacing a toy example with more general conditions on the training data and network widths.

**Strengths And Weaknesses:**

Strengths
1. The paper studies an important question resulting from previous works: when the KKT points are local/global optimum of the max margin problem. I am not aware of any previous work which studies this in detail, so in that sense the contribution of the paper seems to be novel.
2. The provided examples on local optimality of KKT points appear to be simple, and can serve as a good starting point for studying sufficient conditions on the network, the data, and the initialization for proving global max margin in deep learning.
3. The results are clearly presented in the main paper.

Weakness:
1. The negative results are mostly stated for a specific choice of the data points and a toy network (with a few inputs and neurons). This makes me wonder how much of the conclusions in these counterexamples carry over to the more general settings where the data could be drawn from some continuous distribution and the network widths are more general.
2. The paper studies gradient flow as given by an ODE. In several places in the proofs (also in Theorem 2.1), one needs to take time t to infinity. However, it is not very clear why such a solution even exists because an ODE in general can have a finite-time blow up, i.e. the solution has the form [0,t0) where t0 is finite but ||\theta(t)|| may still diverge.

---

> ### Author Response · Authors · 2022-07-31
> **Response**
>
> Thank you for your comments!
>
> Regarding the solution for the ODE:
> Lyu and Li (2019) showed an upper bound for $\lVert \theta(t) \rVert$ (the bound is given by an increasing function in $t$), see Corollary A.11 there. It implies that there is no finite-time blowup.

---

### Official Review · Reviewer_pTzM · 2022-07-07

**Rating:** 7
**Confidence:** 4
**Soundness:** 4 excellent
**Presentation:** 4 excellent
**Contribution:** 4 excellent

**Summary:**

The implicit bias of gradient descent/flow for homogeneous models using losses with an exponential tail has been previously characterized the first-order critical point of a $\ell_2$-max margin problem in parameter space. The purpose of this paper is to investigate whether this critical point can be guaranteed to have stronger properties, such as local or global optimality in the max-margin problem. The authors study a variety of situations including linear or Relu neural networks, of depth 2 or more, with or without weight sharing. In all these situations, they prove whether there is a positive local/global optimality result and if not, they give counter-examples.


**Questions:**

- Thm.3.2. would benefit from a discussion with two related works: [1] and [2]. [1] gives another context where the final margin is suboptimal while [2], on the positive side, shows that there is global margin maximization for infinite-width networks under assumptions (the difference between the two being, it seems, that the limit small initialization/large width are interverted).

[1] Gradient descent on two-layer nets: Margin maximization and simplicity bias, Lyu,  Li,  Wang,  Arora, 2021.

[2] Implicit bias of gradient descent for wide two-layer neural networks trained with the logistic loss, Chizat, Bach, 2020.

**Limitations:**

N/A.

**Strengths And Weaknesses:**


The paper is well written, clear and well organized. It consists in a collection of theoretical results, each comes with a proof strategy in the main text. The various insights are summarized in Table 1 and 2 which, at a glance, give an interesting general picture of the problem. I was quite impressed by the profusion of results and settings considered, and overall I think that the paper will surely be useful for the community of researchers working on the theory of implicit bias of gradient descent.

I don't have particular concerns about the paper (ie "weaknesses") besides the fact that it deals with a rather specific theoretical question which might be only of interest to a restricted number of specialists. But I think that as a community we need this kind of papers that investigate in depth certain theoretical points, so I don't think that this is much of a weakness.

---

> ### Author Response · Authors · 2022-07-31
> **Response**
>
> Thank you for your comments!
>
> Regarding your question:
> A discussion on these papers is given in Appendix A (mostly on Lyu et al.). We may add a discussion in the main paper.

---

### Official Review · Reviewer_fRNg · 2022-07-16

**Rating:** 6
**Confidence:** 4
**Soundness:** 4 excellent
**Presentation:** 3 good
**Contribution:** 3 good

**Summary:**

This paper studies the implicit bias of gradient descent through the lens of margin maximization, and the main focus is to understand what can be guaranteed by the KKT convergence result by Lyu & Li (2019). It then presents a series of positive and negative results on the local and global optimality of KKT points across different settings, depending on whether the neural net is linear/nonlinear, whether it is fully-connected/diagonal, and whether it has shared weights or not. This paper also proposes the notion of per-layer margin maximization to be another path to obtain positive results on the implicit bias, and establishes a local optimality result assuming non-zero inputs to all neurons.

**Questions:**

1. Chizat and Bach(2020); Ji & Telgarsky (2020) prove the global optimality of margin for wide two-layer nets under the assumption that the directions of weight vectors in the first layer approximately cover the unit sphere.  Phuong and Lampert (2020) prove the global optimality of margin on orthogonally separable data. Lyu et al. (2021) prove the global optimality of margin on symmetric and linearly separable data. It would be nice if the authors can reconcile their negative results with these positive results from previous works. For example, how are the assumptions violated in the counter-examples?
2. I'm wondering if the per-layer optimality results (Theorem 5.2 and 5.4) have a close relationship with Corollary 4.5 in Lyu & Li (2019). For linear nets,
\begin{align*}
\left< u^{(l_0)}, \frac{\partial \Phi(\theta'; x_i)}{\partial u^{(l_0)}} \right> = \Phi(\theta'; x_i).
\end{align*}
So Eq. (4) becomes
\begin{align*}
\min_{u^{(l_0)}}  \frac{1}{2} \\|u^{(l_0)}\\|^2 \quad s.t. \quad \forall i \in [n], y_i \left< u^{(l_0)}, \frac{\partial \Phi(\theta'; x_i)}{\partial u^{(l_0)}} \right> \ge 1.
\end{align*}
Note that $(\partial \Phi(\theta'; x_i))/ (\partial u^{(l_0)})$ does not change with $u^{(l_0)}$. So Corollary 4.5 in Lyu & Li (2019) should have implied Theorem 5.2. The ReLU case is similar as $\Phi$ is locally linear.

**Limitations:**

The authors have discussed the limitations of their counter-examples in Lines 83 - 89.
It would be nice if the authors can mention the following limitations which are shared in common with many implicit bias works:
1. The difference between GF and SGD may not be negligible;
2. Reaching a KKT point can take exponential time even for linear logistic regression (Soudry et al., 2018).

**Strengths And Weaknesses:**

#### Strengths:
1. This paper is well-written and presents a nice exhibition of the meaning of Lyu & Li's KKT convergence across different settings.
2. Theorem 3.1 shows that GF on deep linear fully-connected nets converges to the max-margin solution in the parameter space, which complements the previous margin maximization result in the predictor space (Ji & Telgarsky, 2020).
3. The per-layer local optimality result gives a new characterization of the convergence point.

#### Weaknesses:
1. I like the simple counter-examples shown in this paper for margin maximization, but an obvious weakness is that all these examples are for small datasets and small neural nets. It is nice that the authors include discussions like Remarks 3.1 and 3.2 on the generality of the counter-examples, but perhaps the authors could discuss more on large datasets and wide neural nets.
2. It is unclear how the per-layer local optimality result can be connected to generalization or other learning metrics.
3. The negative results are not well contextualized relative to prior works that attempt to prove the global optimality of margin (see Questions below).

#### Minor comments:
* It is not a good idea to put so many important related works in Appendix A. Perhaps the authors could consider to move this section to the main paper.
* It would be nice if the authors can include a section to conclude the paper at the end.

---

> ### Author Response · Authors · 2022-07-31
> **Response**
>
> Thanks for your comments!  We will make revisions accordingly. Below are some specific answers to the comments.
>
> Our negative results help understanding which additional assumptions are required for obtaining positive results. We showed some positive results under certain assumptions, and we agree that establishing more positive results under other assumptions is an intriguing direction for future research.
>
> We agree that it would be helpful to discuss how our counter-examples do not satisfy the assumptions in the prior positive results of Chizat and Bach(2020); Ji & Telgarsky (2020); Phuong and Lampert (2020); Lyu et al. (2021). Indeed, it can be easily verified that the conditions that you mentioned do not hold in any of our negative results.
>
> Moreover, we agree that the claim on the per-layer margin maximization is related to Corollary 4.5 in Lyu and Li (2019), although their Corollary considers an optimization problem w.r.t. all the parameters and we consider a specific layer.

---

> > ### Comment · Reviewer_fRNg · 2022-08-09
> > **Thank you for the response**
> >
> > The paper is still good and I will keep my score unchanged.
> >
> > A minor comment is that Corollary 4.5 in Lyu & Li (2019) can imply Theorem 5.2 (and also the one in the ReLU case) in the strict sense, although their corollary considers an optimization problem w.r.t. all the parameters. This is because one can write down the optimality conditions of the optimization problem in their corollary w.r.t. the parameters in just one single layer, and then use the relationship I mentioned in the review to connect this partial optimality with the per-layer optimality in this paper. This gives an alternative proof for Theorem 5.2, and it would be nice if the authors could mention this connection.

---

### Meta-Review · Area_Chair_Wy68 · 2022-09-05

**Recommendation:** Accept
**Confidence:** Certain

**Metareview:**

The reviewers reached a consensus that the paper can be accepted by NeuRIPS. The AC notices a few weaknesses pointed out by the reviewers and personally thinks the paper's results are somewhat expected. Nevertheless, the AC would like to recommend acceptance.

**Award:**

Yes

---

### Decision · Program_Chairs · 2022-09-14

Accept